# A comparative and experimental study of the reactivity with nitrate radical of two terpenes: α-terpinene and γ-terpinene

Axel Fouqueau[1], Manuela Cirtog[1], Mathieu Cazaunau[1], Edouard Pangui[1], Jean-François Doussin[1], Bénédicte Picquet-Varrault[1]

[1]Laboratoire Interuniversitaire des Systèmes Atmosphériques (LISA), UMR 7583, CNRS, Université Paris-Est-Créteil et Université de Paris, Institut Pierre Simon Laplace (IPSL), Créteil, France

*Correspondance to:* Bénédicte Picquet-Varrault (benedicte.picquet-varrault@lisa.u-pec.fr)

**Abstract.** Biogenic volatile organic compounds (BVOCs) are intensely emitted by forests and crops into the atmosphere. During the night, they react very rapidly with the nitrate radical (NO$_3$), leading to the formation of a variety of functionalized products including organic nitrates and to large amounts of secondary organic aerosols (SOA). Organic nitrates (ONs) have been shown to play a key role in the transport of reactive nitrogen and consequently in the ozone budget, but also to be important components of the total organic aerosol while SOA are known to play a direct and indirect role on the climate. However, the reactivity of BVOCs with NO$_3$ remains poorly studied. The aim of this work is to provide new kinetic and mechanistic data for two monoterpenes (C$_{10}$H$_{16}$), α- and γ-terpinene, through experiments in simulation chambers. These two compounds, which have very similar chemical structures, have been chosen in order to fill the lack of experimental data but also to highlight the influence of the chemical structure on the reactivity.

Rate constants have been measured using both relative and absolute methods. They were found to be $(1.2 \pm 0.5) \times 10^{-10}$ and $(2.9 \pm 1.1) \times 10^{-11}$ cm$^3$ molecule$^{-1}$ s$^{-1}$ for α- and γ-terpinene respectively. Mechanistic studies have also been conducted in order to identify and quantify the main reaction products. Total organic nitrate and SOA yields have been determined. While organic nitrate formation yields appear to be similar, SOA yields exhibit large differences with γ-terpinene being a much more efficient precursor of aerosols. In order to provide explanations for this difference, chemical analysis of the gas phase products were performed at the molecular scale. Detected products allowed proposing chemical mechanisms and providing explanations through peroxy and alkoxy reaction pathways.

## 1 Introduction

Since the early 1980s and the discovery of the nitrate radical (NO$_3$) in the nocturnal troposphere (Noxon et al., 1980; Platt et al., 1980) and stratosphere (Naudet et al., 1981; Noxon et al., 1978), nighttime chemistry is known to be active. NO$_3$ radical is mainly formed by the reaction of nitrogen dioxide (NO$_2$) with ozone and has two very efficient sinks during the day, its photolysis and its reaction with NO (Brown and Stutz, 2012). During the night, but also during the day under low sunlight conditions (e.g. in forest areas), NO$_3$ has been shown to be an efficient oxidant, reacting with a large variety of VOCs including alkenes, aromatics and oxygenated VOCs (Atkinson and

Arey, 2003). For nighttime conditions, reactions of $NO_3$ radical with biogenic volatile organic compounds (BVOCs) are particularly rapid. BVOCs, which include isoprene ($C_5H_8$), monoterpenes ($C_{10}H_{16}$), sesquiterpenes ($C_{15}H_{24}$) but also oxygenated compounds, represent almost 90% of global emissions of VOCs (Guenther et al.,

1995). Most BVOCs have one or several C=C bonds and thus react particularly rapidly with $NO_3$ by addition of the oxidant on the unsaturation(s) leading to lifetimes below the minute for the most reactive ones. This reaction leads to the formation of nitrooxy alkyl radicals which can then evolve into organic nitrates (ONs). Organic nitrates have been shown to act as reservoirs for reactive nitrogen by undergoing long-range transport in the free troposphere before decomposing and releasing NOx in remote regions. They therefore significantly influence the

nitrogenous species (NOy) and ozone budgets in these regions (Ito et al., 2007). Furthermore, some (multifunctional) organic nitrates are low-volatile and highly soluble both in aqueous phase and organic aerosol (Picquet-Varrault et al., 2019) and are thus capable of strongly partitioning to the atmospheric condensed phases (droplets, aerosols). Recent field observations of the aerosol chemical composition have shown that organic nitrates ranged from 10 to 75% of total organic aerosol (OA) mass (Kiendler-Scharr et al., 2016; Lee et al., 2016;

Xu et al., 2015) suggesting that these species are important components of OA. They can thus significantly affect the aerosols physical and chemical properties, and particularly their optical and hygroscopic properties, controlling their direct and indirect impacts on climate. A good understanding of the reactions BVOCs + $NO_3$ is thus necessary. Nevertheless this chemistry remains poorly studied and except for isoprene, α- and β-pinene, which have been largely studied, it is thinly understood so far.

Among the most occurring terpenes (Geron et al., 2000), α-terpinene and γ-terpinene have been detected in many tree emissions. For example, α-terpinene has been shown to represent 12 % of subalpine fir monoterpenes emissions and γ-terpinene up to 19 % of Sequoia sempervirens' ones (Geron et al., 2000). These two molecules have very similar structures, the only difference being the position of the double bonds (see Fig. 1), which are conjugated for α-terpinene and not for γ-terpinene. Their reactions with $NO_3$ have been subject to few studies. For

both compounds, rate constants have been measured in two studies: γ-terpinene has been the object of absolute and relative rate determinations (Martínez et al. 1999 and Atkinson et al. 1985 respectively) which are in good agreement within uncertainties. For α-terpinene, the rate constant was measured only by relative rate studies using the same reference compound and the two values differ by 80% (Atkinson et al. 1985 and Berndt, Kind, and Karbach 1998). This compound was shown to be very reactive (with rate constant around $10^{-10}$ cm$^3$ molecule$^{-1}$ s$^{-1}$

) making technically difficult an absolute determination. The mechanism for the oxidation of γ-terpinene by $NO_3$ radical has been investigated by only one study (Slade et al., 2017) and no mechanistic study has ever been published for α-terpinene to our knowledge. In the study of Slade et al., 2017, total organic nitrate and SOA yields were determined and some products were detected at the molecular scale allowing proposing a mechanism. New kinetic and mechanistic studies are therefore necessary to better assess the impact of these processes on air quality

and radiative forcing.

The aim of this work is to provide new kinetic and mechanistic data for the reactions of α- and γ-terpinene with $NO_3$ radical using atmospheric simulation chambers. To provide precise and accurate rate constants, absolute rate determinations are conducted for both compounds, using an Incoherent Broad Band – Cavity Enhanced Absorption

Spectroscopy (IBBCEAS) technique recently interfaced to the CSA chamber at LISA (Fouqueau et al., 2020). This technique allows *in situ* monitoring of $NO_3$ radicals with a very good time resolution (10 s) and at very low

concentration (ppt level), these two features being mandatory for kinetic study of fast reactions. For γ-terpinene, a relative rate determination is also performed. In addition, mechanistic studies have been performed for both compounds by providing total organic nitrate and SOA yields and identification of individual gas phase products. It allows proposing reaction mechanisms for the two compounds. For γ-terpinene, these results are compared to those obtained by the previous study. Finally, differences observed for α- and γ-terpinene in kinetic and mechanistic data are discussed in regards to the chemical structures of the two compounds.

## 2 Experimental section

### 2.1 Chemicals

α- and γ-terpinene were purchased from Sigma-Aldrich Co. at respectively 85% and 97% of purity. For both terpenes, a purification stage has been conducted in a vacuum line prior to their injection into the chamber. This purification is expected to remove high volatility impurities. For low volatility impurities, it is expected that they will remain in the sample (condensed phase). However, as the impurities remain unknown, we cannot state with certainty that this purification is 100% efficient and it should be considered that it may generate additional uncertainty on the product yields, in particular for α-terpinene. $NO_3$ radicals were generated *in situ*, by thermal dissociation of $N_2O_5$ (Eq. (1)), previously synthesized in a vacuum line by the reaction between $O_3$ and $NO_2$ (Eq. (2) and Eq. (3)) adapted from (Atkinson et al., 1984a; Schott and Davidson, 1958). The detailed protocol is presented in Picquet-Varrault et al., 2009.

$$N_2O_5 + M \rightleftarrows NO_3 + NO_2 + M \tag{1}$$
$$O_3 + NO_2 \rightarrow NO_3 + O_2 \tag{2}$$
$$NO_3 + NO_2 + M \rightarrow N_2O_5 + M \tag{3}$$

### 2.2 Chamber facilities and analytical devices

Experiments were conducted in two different simulation chambers: CSA and CESAM chambers. The CSA chamber was used for kinetic experiments. It is made of a 6 meters long Pyrex® reactor having a volume of 977 L (Doussin et al., 1997) and being equipped with a homogenization system that allows a mixing time below one minute. This chamber is dedicated to gas phase studies and is hence equipped with several analytical devices for gas phase monitoring. An *in situ* multiple reflection optical system coupled to a FTIR (Bruker Vertex 80) spectrometer allows monitoring organic species in the chamber. Infrared spectra were recorded with a resolution of 0.5 cm$^{-1}$, an optical path length of 204 m and a spectral range of 700-4000 cm$^{-1}$. Integrated band intensities (in cm molecule$^{-1}$, logarithm base e) used to quantify the species are: $IBI_{\alpha\text{-terpinene}}$ (790-850 cm$^{-1}$) = (1.9 ± 0.2) × 10$^{-18}$, $IBI_{\gamma\text{-terpinene}}$ (920-990 cm$^{-1}$) = (1.01 ± 0.02) × 10$^{-18}$, $IBI_{NO2}$ (1530-1680 cm$^{-1}$) = (5.6 ± 0.2) × 10$^{-17}$, $IBI_{HNO3}$ (840-930 cm$^{-1}$) = (2.1 ± 0.2) × 10$^{-17}$, $IBI_{N2O5}$ (1205-1275 cm$^{-1}$) = (1.7 ± 0.1) × 10$^{-17}$. This technique also allowed measuring the total organic nitrate concentration by considering that all these species absorb at 1250 cm$^{-1}$ and 850 cm$^{-1}$ which correspond to absorptions of –ONO$_2$ function and by assuming that their absorption cross sections are similar whatever the organic nitrate considered. This hypothesis was verified in our research group with the analysis of standards. In this study, we used $IBI_{ON}$ (900-820 cm$^{-1}$) = (9.5 ± 2.9) × 10$^{-18}$ cm molecule$^{-1}$. Uncertainties on concentrations of species measured by FTIR include the uncertainty on the IBIs and the uncertainty on the spectra analysis.

For absolute rate determination, $NO_3$ was monitored from its visible absorption at 662 nm with an *in situ* IBBCEAS technique which has recently been coupled to the CSA. It is described in details in Fouqueau et al., 2020a. This technique also allows $NO_2$ monitoring. Having a precise knowledge of the wavelength dependent mirror reflectivity, $R(\lambda)$, is one of the most critical point of the IBBCEAS technique. It was therefore determined prior to
each experiment by introducing a known amount of $NO_2$ (several hundred ppb) into the chamber. The cross sections used to quantify $NO_2$ are from (Vandaele et al., 1997) and those used for $NO_3$ quantification are from Orphal et al., 2003. At 662.1 nm, which corresponds to the $NO_3$ maximum absorption, the cross section is $(2.13 \pm 0.06) \times 10^{-17}$ $cm^2$ molecule$^{-1}$. Thanks to the very high reflectivity of mirrors (99.974 ± 0.002%), the optimum optical path length was found to be 2.5 km leading to a detection limit of 6 ppt for 10 s of integration time. The
uncertainty on $NO_3$ concentrations by IBBCEAS was estimated to be 9%, with a minimum absolute value of 3 ppt (Fouqueau et al., 2020). This uncertainty includes the uncertainties on the reflectivity of the mirrors, the $NO_3$ absorption cross sections, and the data treatment.

Finally, in order to monitor organic reactants and products, a high resolution PTR-ToF-MS (Kore Series 2e, mass resolution of 4000) was used in two ionization modes, $H_3O^+$ and $NO^+$. When used in standard operational
conditions, i.e. with $H_3O^+$ ionization, it has been shown that organic nitrates are subject to important fragmentation (Müller et al. 2012 ; Aoki, Inomata, and Tanimoto 2007). In order to reduce this fragmentation, Duncianu et al. 2017 has adapted the instrument operating procedure for organic nitrate detection by reducing the electric field in the reactor. The same study has also developed a $NO^+$ ionization mode, by replacing the ionization gas (water vapor) with dry air and also by applying a reduced electric field. These two modes allow cross checking for the
identification of the products. This method was characterized and validated thanks to experiments with various standards of organic nitrates (alkyl nitrates, carbonyl nitrates and hydroxynitrates) allowing the authors to propose ionization patterns for each type of organic nitrates and for both ionization modes. Hence, in $NO^+$ ionization mode, organic nitrates were shown to be ionized by charge transfer or by an $NO^+$ adduct formation and therefore to be detected at their own mass (M) or at M+30. Hydroxynitrates have been detected at M-1 suggesting that the
ionization proceeds mainly by a hydrogen loss.

The second simulation chamber is CESAM chamber (Experimental Chamber of Multiphase Atmospheric Simulation) (Wang et al., 2011). It has been specifically designed to study multiphase processes. In this work, it was used to investigate the mechanisms and the SOA formation. CESAM is a 4177 liters stainless-steel evacuable reactor and is equipped with a fan that allows mixing time of approximatively one minute. Aerosol lifetimes in
CESAM being very long (up to 4 days), it is particularly suited for SOA studies. This chamber is equipped with dedicated analytical instruments for gas and aerosol phases. It is coupled with an *in situ* long path FTIR spectrometer (Bruker Tensor 37) allowing acquiring spectra in the 700-4000 cm$^{-1}$ spectral range with a resolution of 0.5 cm$^{-1}$ and an optical path of 174.5 m. It is also equipped with a PTR-ToF-MS which was operated in both $NO^+$ and $H_3O^+$ ionization modes. The size distribution of the particle phase was measured with a Scanning Mobility
Particle Sizer (SMPS) composed of a TSI Classifier model 3080 and Differential Mobility Analyzer (DMA) model 3081 coupled to a Condensation Particle Counter (CPC) TSI model 3772 which allowed measurements in the range of 20-880 nm. To convert size distribution into mass distribution, a particle density of 1.4 g cm$^{-3}$ was used, estimated to be the density of SOA formed by BVOC+$NO_3$ reactions (Fry et al. 2014; Draper et al. 2015; Boyd et al. 2015). Some experiments conducted on α-terpinene lead to the formation of large particles (diameters > 800

nm). In this case, a Palas Welas (Welas digital 2000) was used in addition to the SMPS, to measure the particle size distribution. This instrument is based on an optical measurement and allows covering a wider size range (0.2-17 µm).

During experiments, filter sampling was proceeded, allowing the measurement of total ONs yield in the aerosol phase. The filter analysis was performed by FTIR after extraction of particles in liquid phase, following a protocol described by Rindelaub et al., 2015: SOA are extracted in 5 mL of $CCl_4$. Organic nitrates were quantified from standard solutions of 2 types of organic nitrates (nitrooxypropanol and *tert*-butyl nitrate). IBIs for the two standards were 510 and 580 L $mol^{-1}$ $cm^{-2}$ respectively. The difference between the two of them is small, so the integrated absorption cross section of organic nitrates in liquid phase was considered to be: $IBI_{ONs}$ (1264-1310 $cm^{-1}$) = 557 ± 110 L $mol^{-1}$ $cm^{-2}$.

The pressure into the chamber was maintained constant by introducing synthetic air in order to compensate the decrease of pressure due to instrument sampling. This leads to a weak dilution of the mixture, here less than 20% for an experiment length of 3 hours. All data presented in the following sections were corrected by dilution and, for the SOA measurements, by the particles wall losses, which have been characterized in CESAM: wall loss rates were determined as a function of the diameter of the particles and interpolated using the (Lai and Nazaroff, 2000) model (friction velocity u*=3.7 cm $s^{-1}$ – see Lamkaddam et al., 2017). Due to the material used for the walls (stainless steel), this correction has been found to be small in the present case.

### 2.3 Kinetic study

Kinetic experiments were performed in the CSA chamber at room temperature and atmospheric pressure, in a mixture of $N_2/O_2$ (generated using 80 % of $N_2$ from liquid nitrogen evaporation, purity > 99.995 %, $H_2O$ < 5 ppm, Messer and 20 % of $O_2$, quality N5.0, purity > 99.995 %, $H_2O$ < 5ppm, Air Liquide). Rate constants were determined by using both relative and absolute rate methods in order to provide accurate kinetic data. During a typical experiment, organic reactants are introduced into the chamber and left in the dark for a period of approximatively one hour in order to check that there is no significant loss of the compounds. Then, $NO_3$ was generated *in situ* by thermal decomposition of $N_2O_5$ (see section 2.1) which was introduced into the chamber using several stepwise injections for a complete consumption of the BVOC.

For absolute rate determinations, concentrations of BVOC and $NO_3$ radical were monitored by PTR-ToF-MS and IBBCEAS respectively. Low mixing ratios of BVOC (between 15 and 50 ppb) have been used in order to reduce the SOA formation which leads to a significant decrease of the IBBCEAS signal due to light absorption/scattering and mirror soiling by particles. In order to allow monitoring fast decay of reactants, low integration time (10 s) was used for both techniques. In addition, prior to each experiment, several hundred ppb of $NO_2$ (between 450 and 650 ppb) were introduced to i) measure the reflectivity of the IBBCEAS mirrors and ii) shift the equilibrium between $N_2O_5$, $NO_3$ and $NO_2$, and hence slowing down the $N_2O_5$ decomposition and the BVOC + $NO_3$ reaction.

$$BVOC + NO_3 \rightarrow Products \tag{4}$$

Then, the second-order kinetic equation is:

$$-\frac{d[BVOC]}{dt} = k_{BVOC}[BVOC][NO_3]$$    (5)

This equation can be approximated for small time intervals:

$$-\Delta[BVOC] = k_{BVOC}[BVOC][NO_3]\Delta t$$    (6)

where $-\Delta[BVOC]$ corresponds to the consumption of BVOC during the time interval $\Delta t$ and $[BVOC]$ and $[NO_3]$

are averaged concentrations during this interval. By plotting $-\Delta[BVOC]$ vs $[BVOC] \times [NO_3] \times \Delta t$, a straight line

with a slope corresponding to $k_{BVOC}$ is obtained. It should be mentioned that the determination of the rate constant

is thus not affected by losses of $NO_3$ radicals due to reaction with other species (products, $RO_2$ radicals …).

Uncertainty on the rate constant was calculated by considering twice the standard deviation on the slope.

For relative rate determination, the decay of the BVOC was monitored relatively to a reference compound using

PTR-ToF-MS and FTIR techniques. If it is assumed that reaction with $NO_3$ is the only fate of both the studied

compound (BVOC) and the reference compound (Ref.), and that neither of these compounds is reformed at any

stage during the experiment, it can be shown that (Atkinson, 1986):

$$ln\left(\frac{[BVOC]_{t_0}}{[BVOC]_t}\right) = \frac{k_{BVOC}}{k_{Ref.}} ln\left(\frac{[Ref.]_{t_0}}{[Ref.]_t}\right)$$    (7)

where $[BVOC]_{t_0}$ and $[Ref.]_{t_0}$ are the concentrations of BVOC and Ref. at time $t_0$ (before the beginning of the

oxidation), $[BVOC]_t$ and $[Ref.]_t$ are the concentrations at time t and $k_{BVOC}$ and $k_{Ref.}$ are the rate constants with

$NO_3$ respectively.

In this work, two different reference compounds with well-known rate constants were used: 2,3-dimethyl-2-butene

and 2-methyl-2-butene. Due to the lack of recommendation by IUPAC for the reaction between these compounds

and $NO_3$ radical, rate constants were calculated as the mean values of the determinations available in the literature

(Atkinson, 1988; Atkinson et al., 1984; Atkinson et al., 1984; Benter et al., 1992; Berndt et al., 1998). The

uncertainties on reference rate constants were calculated as twice the standard deviation of all the values. The

obtained rate constants are: $k_{2,3\text{-dimethyl-2-butene}} = (5.5 \pm 1.7) \times 10^{-11}$ cm$^3$ molecule$^{-1}$ s$^{-1}$ and $k_{2\text{-methyl-2-butene}} = (9.6 \pm 1.6)$

$\times 10^{-12}$ cm$^3$ molecule$^{-1}$ s$^{-1}$. Finally, the uncertainty on $k_{BVOC}$ was calculated by considering the relative uncertainty

corresponding to the statistical error on the linear regression ($2\sigma$) and the error on the Ref. rate constant.


**2.4 Mechanistic study**

Mechanistic experiments were performed in CESAM chamber at room temperature and atmospheric pressure, in

the same mixture of $N_2/O_2$ (80/20) as the kinetic experiments. During a typical experiment, the BVOC is introduced

into the chamber and left in the dark to estimate potential wall losses. No significant wall loss was observed ($k_d <$

$10^{-7}$ s$^{-1}$). Then $N_2O_5$ is introduced. Two methods were used to inject $N_2O_5$ in order to optimize the decay rate of

the BVOC: by stepwise injections and by slow continuous injections. The second method was observed to be more

efficient to slow down the oxidation, and thus to better control the SOA formation. BVOC and gas phase products

were monitored by both PTR-ToF-MS and FTIR spectrometer. Some experiments were conducted with two PTR-

ToF-MS, allowing analyzing the gas phase in both $NO^+$ and $H_3O^+$ ionization mode simultaneously. When both

instruments were not available at the same time, experiments were duplicated to allow the product detection with

the two ionization modes. Production of SOA was monitored by SMPS. No seed particles were introduced in order to determine SOA yields under low aerosol content. Sampling on filters were performed for high concentration experiments (>150 ppb) in order to determine the total organic nitrate concentration in the particle phase (see section 2.2). The sampling on filters started at the end of the oxidation (when the precursor has completely reacted) and lasted between 3 and 6 hours. A charcoal denuder was used to remove organic compounds from gas phase.

Total organic nitrate yields in gas phase were determined by plotting the molecular concentration of organic nitrates as a function of the molecular concentration of the BVOC reacted and by calculating the slope of the straight line. Organic nitrate yields in SOA phase were calculated by measuring the final concentration of organic nitrates and by dividing it by the total reacted BVOC concentration for each experiment. Uncertainty on the yield was calculated as the sum of the relative uncertainties on organic nitrates and BVOC concentrations.

The SOA yield is defined as the ratio of the mass concentration of SOA produced, $M_0$, divided by the mass concentration of the BVOC reacted, $\Delta$BVOC. For all experiments, the SOA yield was calculated for each data point, but also once the BVOC has been totally consumed, hence providing both time-dependent and overall SOA yields. These yields were plotted as a function of the organic aerosol mass and fitted by a two-product model described by Odum et al., 1996:

$$Y = M_0 \left[ \frac{\alpha_1 K_{p,1}}{1+K_{p,1}M_0} + \frac{\alpha_2 K_{p,2}}{1+K_{p,2}M_0} \right] \tag{8}$$

Where $\alpha_1$, $\alpha_2$ and $K_{p,1}, K_{p,2}$ are stoichiometric factors and partitioning coefficients (in m³ µg⁻¹) of the two hypothetical products respectively. Due to the slow injections of $N_2O_5$, SOA equilibrium was expected to be reached at small time steps and time-dependent yields have been used. This also allowed obtaining yields for small aerosol content in the chamber.

As described in section 2.2, oxidation products were detected thanks to PTR-ToF-MS measurements in two ionization modes ($H_3O^+$ and $NO^+$). However, quantification of these products was not performed due to the lack of standards. Finally, vapor pressures $P^{vap}$ have been estimated using SIMPOL-1 method of Pankow and Asher, 2008 in order to evaluate their contribution to SOA formation via the GECKO-A website (http://geckoa.lisa.u-pec.fr/generateur_form.php, last access May 12th 2020). Raoult's law (Valorso et al., 2011) has been used to estimate also the fraction of a product i in the condensed phase $\xi_{aer}^i$:

$$\xi_{aer}^i = \frac{N_{i,aer}}{N_{i,aer}+N_{i,gas}} = \frac{1}{1+\frac{\overline{M_{aer}}\gamma_i P_i^{vap}}{C_{aer}RT}\times 10^6} \tag{9}$$

where $N_{i,gas}$ and $N_{i,aer}$ are the concentrations (in molecule cm⁻³) of the product i in the gas and particle phases respectively, $\overline{M_{aer}}$ is the mean molecular weight of SOA species (g mol⁻¹), $C_{aer}$ is the total SOA mass concentration (µg m⁻³), R is the gas constant (atm m³ K⁻¹ mol⁻¹), T the temperature (K), $P_i^{vap}$ is the vapor pressure and $\gamma_i$ is the activity coefficient of product i (in this study, $\gamma_i = 1$). The mean molecular weight has been estimated to be the mean value for low volatility products which were detected.

The calculation of $\xi_{aer}^i$ is highly dependent of the estimated vapor pressure. Pankow and Asher, 2008 showed that SIMPOL-1 technique allows predicting $P^{vap}$ with an error between 50 % and 60 % for $P^{vap} < 10^{-6}$ atm. $\xi_{aer}^i$ is

therefore associated with a high uncertainty and can only be used as an indicator. $\xi_{aer}^i$ can also be compared to partitioning coefficients $K_p$ in Eq. (8), using the following equation:

$$K_p = \frac{N_{i,aer}}{N_{i,gas}} \times \frac{1}{C_{aer}} = \frac{\xi_{aer}^i}{1-\xi_{aer}^i} \times \frac{1}{C_{aer}} \tag{10}$$

## 3 Results

### 265   3.1 Kinetic results

The list of kinetic experiments and their corresponding experimental conditions are presented in Table 1. Absolute rate determinations were conducted for α- and γ-terpinene while relative rate one was performed only for γ-terpinene. For each method, between three and five experiments were conducted.

Kinetic results obtained for γ-terpinene by relative rate method are presented in Fig. 2. Good linear tendencies are
observed for the two reference compounds and data obtained using PTR-Tof-MS and FTIR measurements are in good agreement. Linear regressions have been first performed for the both individual data sets (PTR-ToF-MS and FTIR). Because the results were in good agreement for both of the measurement techniques, linear regression was applied to all the values (by mixing data sets), leading to $k_{\gamma\text{-terpinene}} = (3.0 \pm 1.1) \times 10^{-11}$ cm$^3$ molecule$^{-1}$ s$^{-1}$ with 2,3-dimethyl-2-butene and $(2.7 \pm 0.6) \times 10^{-11}$ cm$^3$ molecule$^{-1}$ s$^{-1}$ with 2-methyl-2-butene. It can be concluded that rate
constants obtained with the two references are in very good agreement.

For absolute rate determinations, typical time profiles of reactants measured with PTR-ToF-MS, FTIR and IBBCEAS are presented in Fig. S1 for the experiment of 2017/01/30 on γ-terpinene. Good agreement is observed for γ-terpinene and between FTIR and PTR-ToF-MS data. Good agreement is also observed for $NO_2$ between FTIR and IBBCEAS data, with an exception for the first experimental point for $NO_2$ following the injection of
$N_2O_5$ for which a good mixing is probably not fully achieved yet. These agreements are particularly satisfying considering the fact that the two instruments do not sample in the same volume of the chamber: FTIR provides an integrated measurement of the absorbing species over the whole length of the chamber, IBBCEAS provides an integrated measurement in the width of the chamber and PTR-ToF-MS samples the mixture in one point. This comparison demonstrates that the mixing of the chamber is efficient enough to allow combining data from different
instruments for absolute rate determination.

Kinetic plots for absolute kinetic determinations gathering results from all experiments are presented in Fig. 3 for both BVOC. As explained above, the first experimental point following the injection of the reactants was not taken into account. Due to the low integration time used for both measurement techniques, a relatively high noise has been observed for BVOC and $NO_3$ concentrations. Kinetic results are thus subject to relatively high uncertainties.
Rate constants measured by the absolute rate method are $(3.0 \pm 0.9) \times 10^{-11}$ cm$^3$ molecule$^{-1}$ s$^{-1}$ for γ-terpinene and $(1.2 \pm 0.3) \times 10^{-10}$ cm$^3$ molecule$^{-1}$ s$^{-1}$ for α-terpinene.

The absolute values are compared to those obtained by the relative method (for γ-terpinene) and to those already published in the literature in Table 2. It should be noticed that relative rate determinations from Atkinson et al. 1985 and Berndt et al. 1996 have been updated by using the same reference rate constants as the one used for this

study (see section 2.3). Uncertainties on the reference rate constants have also been added to the statistical errors provided by the authors.

For γ-terpinene, the three rate constants obtained by this study, i.e. absolute and relative determinations with two reference compounds, are in very good agreement. They are also in good agreement with the relative kinetic study of Atkinson et al. 1985 and with the absolute study of Martinez et al. 1999, even if the second one appears to be

20 % lower. For α-terpinene, the absolute rate determination provided by this study has been compared to the previous relative determinations provided by Atkinson et al., 1985 and by Berndt et al., 1996. These two relative rate studies were performed with the same reference compound but using two different experimental setups: a flow reactor (Berndt et al., 1996) and a simulation chamber (Atkinson et al., 1985). When considering the overall uncertainties on these rate constants (approx. 40 %) which include the uncertainty on the reference rate constant,

the data seem to be in agreement but when comparing the ratio $k_{BVOC}/k_{ref}$, it appears that they are in fact, not congruent. No explanation was provided by the authors to explain this disagreement. However, for comparison with absolute rate determination, the overall uncertainty had to be considered. Within uncertainties, the value provided here is in agreement with these two previous. In conclusion, this study allows providing new kinetic data for α- and γ-terpinene and confirming the values obtained by the few previous studies. It also provides the first

absolute rate determination for α-terpinene.

When comparing the reactivity of the two terpenes, it can be seen that α-terpinene is much more reactive than γ-terpinene (by a factor of approximatively four) and this can easily be explained by the conjugation of the double bonds for α-terpinene. Indeed, after the addition of $NO_3$ radical on one of the double bonds, the alkyl radical formed is stabilized by the delocalization of the single electron. Experimental data have also been compared to

rate constants estimated by the structure-activity relationship (SAR) developed by Kerdouci et al., 2014 for the reaction between BVOCs and $NO_3$ (see Table 2). This SAR has been shown to estimate rate constants within a factor of 2. By taking into account these uncertainties, it can be considered that experimental and estimated rate constants are in good agreement. In particular, the significant increase of the rate constant due to the conjugation of the double bonds is well reproduced by the SAR.

**4 Mechanistic results**

Eleven mechanistic experiments were conducted in CESAM chamber for γ-terpinene and eight for α-terpinene, during which the formation of gas phase products and SOA was monitored. Experimental conditions as well as organic nitrate and SOA yields obtained for all experiments are presented in Table 3. Figure 4 presents, as an example, time-profiles of reactants and products (after correction from dilution) for the experiment of 2017/25/04

on γ-terpinene. During this experiment, $N_2O_5$ was introduced into the chamber by slow continuous injection (shown by the red area) in order to ensure a progressive consumption of the BVOC. This injection leads to the formation of large amounts of $NO_2$ and $HNO_3$ which can respectively be explained by $N_2O_5$ decomposition and hydrolysis on surfaces (lines, chamber walls). It should also be noticed that because $N_2O_5$ was introduced continuously in small amount, its concentration remains below the detection limit as long as the concentration of

BVOC remains high. γ-terpinene is totally consumed within approximatively 30 min and this reaction leads to the formation of large amounts of organic nitrates and SOA. Starting from approximatively 500 ppb of γ-terpinene, more than 200 ppb of organic nitrates and 800 µg/m$^3$ of aerosol are formed. The aerosol size distribution presented

in Fig. 4 shows that particles have mean diameters around 300-400 nm. PTR-ToF-MS signals (m/z) are presented in Fig. S2. Several masses corresponding to oxidation products have been detected with the most intense signals being for m/z = 115, 169 and 153. Time profiles and identification of these signals are discussed later.

**4.1 SOA yields**

Time-dependent and overall SOA yields ($Y_{SOA}$) for both compounds have been plotted as a function of the aerosol mass ($M_0$) in Fig. 5. A two products model, defined by Odum (Odum et al., 1996), see Section 2. 4, has been applied for the two curves. For each compound, experimental points obtained from different experiments are in fairly good agreement and show similar tendencies. Yields obtained for γ-terpinene can reach 40 % whereas they are below 2% for α-terpinene. In the case of α-terpinene this value is subject to possible slight underestimation due to the low purity of the α-terpinene sample (see section 2.1). These results demonstrate that γ-terpinene is a very efficient SOA precursor which is not the case for α-terpinene.

As shown in Fig. 5, fitted plots appear to be well constrained for small aerosol contents (below 50 µg m$^{-3}$) due to many points measured in this area. This is a consequence of the slow injection of $N_2O_5$ which allows a progressive BVOC oxidation. Fitted parameters have been found to be: $\alpha_1 = 0.22$ ; $K_{p,1} = 3.4 \times 10^{-3}$ m$^3$ µg$^{-1}$ and $\alpha_2 = 0.22$ ; $K_{p,2} = 4.5 \times 10^{-2}$ m$^3$ µg$^{-1}$ for γ-terpinene and $\alpha_1 = 0.01, K_{p,1} = 4.5 \times 10^{-1}$ m$^3$ µg$^{-1}$ and $\alpha_2 = 0.01, K_{p,2} = 3.5 \times 10^{-2}$ m$^3$ µg$^{-1}$ for α-terpinene. For both terpenes, SOA production can be successfully modeled by two classes of products having similar stoichiometric factors but different volatilities ($K_{p,1}$ and $K_{p,2}$ differ by more than a factor 10). However, for γ-terpinene, stoichiometric factors are two order of magnitude higher than those obtained for α-terpinene, leading to higher SOA yields. For α-terpinene, one class of products appears to have very low volatility ($K_{p,1} = 4.5 \times 10^{-1}$ m$^3$ µg$^{-1}$ ) but is formed with very low yield. One can estimate the uncertainties of these parameters by looking at the fit sensitivity. It appears to be very sensitive to $\alpha$ (with an associated error estimated to 5 %) and less to $K_p$ (with an error estimated to 50 %).

For α-terpinene, our study provides the first determination of SOA yields. For γ-terpinene, SOA yields have been compared with those provided by Slade et al., 2017. This study used seeds for half of the experiments, and did not observe significantly different yields for experiments conducted with and without seeds. In the study of Slade et al., 2017, a density of 1.7 g cm$^{-3}$ (which corresponds to the density of seed particles) was used to convert aerosol volume into mass. To allow comparison, data obtained by Slade et al. have been corrected in order to use the same density as the one used here, i.e. 1.4 g cm$^{-3}$ (Fry et al., 2014). Odum curves have been plotted and compared to our results: up to 200 µg m$^{-3}$, both Odum curves follow a similar tendency. At higher concentrations, yields measured by Slade et al., 2017 are significantly higher than those obtained here. Despite the use of a progressive $N_2O_5$ injection by Slade et al., 2017, reactions were fast and the BVOC was totally consumed within approximatively 15 minutes. In our experiments the oxidation time ranged between 15 (punctual injection of $N_2O_5$) and 70 min (longest continuous injection of $N_2O_5$) and we observed that faster oxidations result in higher SOA yields obtained by locally generation of high concentrations of semi-volatile species. Hence, the experiment of the 2017/02/15 which is one of the fastest continuous injection experiments (21 minutes) appears to be more congruent with the results from Slade et al., 2017.

In conclusion, both compounds appear to have very different behavior towards SOA production: for an ambient aerosol mass loading of 10 µg m$^{-3}$, which is typical of biogenic SOA impacted environments (Slade et al. 2017), yields of 10 % and 1 % have been found respectively for γ-terpinene and α-terpinene. For higher aerosol mass loading observed in polluted atmospheres (between 500 and 1000 µg m$^{-3}$), yields can reach 30-40 % for γ-terpinene and only 2 % for α-terpinene.

**4.2 Organic nitrates yields**

The formation yields of total organic nitrates in the gas phase ($Y_{ONg}$), have been investigated by plotting their concentration as a function of the BVOC consumption for both α- and γ-terpinene (see Fig. 6). The linearity of these plots and the fact that the slopes at the origin are different from zero indicate that i) organic nitrates are primary products and ii) if primary organic nitrates are subject to loss processes, e.g. through reaction with NO$_3$ radicals, they may produce secondary organic nitrates, so that the ON yield is constant. It is also expected that organic nitrates adsorb on the stainless steel walls. Indeed, the loss rates of several multifunctional organic nitrates (in particular carbonyl-nitrates) have been observed in previous studies (Suarez-Bertoa et al., 2012; Picquet-Varrault et al., 2020) and were found to range between 0.5 and $2 \times 10^{-5}$ s$^{-1}$. However, as yields of organic nitrates in the gas-phase were calculated during a relatively short period (less than 1 hour), these wall losses are expected to be low (less than 10%) and this is confirmed by the good linearity of the plots. Molar $Y_{ONg}$ obtained for both BVOCs are very similar: 47 ± 10 % for γ-terpinene and 43 ± 10 % for α-terpinene. These results confirm that organic nitrates are major products of BVOC+NO$_3$ reactions. For γ-terpinene, the yield obtained here has been compared to the only value previously reported in the literature by Slade et al. 2017: 11 ± 1%. Despite the fact that experimental conditions are very similar, $Y_{ONg}$ differ by a factor of four. Loss reactions of organic nitrates in particle phase that shift the partitioning equilibrium are invoked by the authors to explain the surprisingly low obtained yield. No influence of RH on organic nitrate yields has been noticed. Another suggested hypothesis advanced by the authors is an epoxidation of hydroxynitrates in particle phase, followed by a loss of the NO$_2$ group. However, it is expected that these reactions also occur in our experiments and this hypothesis cannot explain the differences observed between the two studies. Concerning α-terpinene, our study provides the first organic nitrate yield.

$Y_{ONg}$ obtained in this study have also been compared to those obtained for other terpenes available in the literature. Except for α-pinene, for which yields vary between 10 and 30 % (Fry et al., 2014; Hallquist et al., 1999; Spittler et al., 2006), those obtained for isoprene and monoterpenes are close to our study, with yields higher than 30 %. For β-pinene for example, they vary between 40 and 74 % (Hallquist et al. 1999; Fry et al. 2014; Boyd et al. 2015) and for limonene, between 30 and 72 % (Fry et al., 2014, 2011; Hallquist et al., 1999; Spittler et al., 2006).

Considering the fact that organic nitrates may partition into/onto aerosols, yields of total organic nitrates in the particle phase ($Y_{ONp}$) have also been measured by collecting particles on filters and analyzing them by FTIR. Values are presented in Table 3. For γ-terpinene, molar yields range between 1 and 8 %, presenting a high dispersion and for α-terpinene range between 1 and 3 %. This dispersion can be explained by the fact that i) $Y_{ONp}$ directly depends on the total SOA mass concentration, and consequently on the concentration of reacted BVOC and ii) the concentrations of ON are low (minimum is $5 \times 10^{-5}$ mol L$^{-1}$) and thus subject to high variability. The first correlation appears to be clear in Table 3: for high concentration experiments (~400 ppb), yields reach 10 %,

and for low concentration ones, yields appear to be between 2 and 3 %. It should also be mentioned that these yields may be affected by wall losses of organic nitrates. Considering that these have been measured by collecting particles several hours after the beginning of the experiment, this may lead to a non-negligible loss of organic

nitrates (up to 25%), and therefore to an underestimation of the yields. Because wall loss rates can vary from an experiment to another one, this can also explain the variability of $Y_{ONp}$. Also, in the case of α-terpinene the low purity of the sample may lead to a slight underestimation of the calculated yields in both particle and gas phase (see section 2.1).

The molar $Y_{ONp}$ for γ-terpinene are in good agreement with the one provided by Slade et al., 2017 (3(+2/-1) %)

using the same analytical method. The values presented by Slade et al., 2017 present also a high dispersion similar to the one presented here.

In order to estimate the fraction of organic nitrates in SOA, ON yields in the aerosol phase have been compared to SOA yields, both expressed in mass. Thus, molar $Y_{ONp}$ have been converted in mass yields by considering a unique molecular weight which is representative of the expected oxidation products. Here, a hydroxynitrate having the

molecular formula $C_{10}H_{17}O_4N$ and the molecular weight of 215 g mol$^{-1}$ has been used. As discussed later, this compound has been detected as a product by PTR-ToF-MS. It is clear that this assumption generates large uncertainty of the ON mass yield, especially if products with much higher molecular weight are formed, for example by polymerization in condensed phase. However, since organic nitrates were not quantified individually, this method allows estimating the contribution of organic nitrates to SOA. The values obtained are presented in

the Table 3. For γ-terpinene, the mass yields have been found to range between 3 and 13 %, and for α-terpinene, between 3 and 6 %. By comparison with SOA yields, it is estimated that organic nitrates represent ~50 % of the SOA for γ-terpinene and ~100 % for α-terpinene. Organic nitrates are therefore major components of the SOA produced by the $NO_3$ oxidation of these two BVOCs.

This conclusion can be compared with some fields studies results (Kiendler-Scharr et al., 2016; Ng et al., 2017)

that found that organic nitrates are a major component of organic aerosol, with a proportion that can reach almost 80 %. Even if organic nitrates can be formed by other chemistries, studies showed that an enhancement of organic nitrates in SOA has been observed in $NO_3$ radical impacted regions: during night (Gómez-González et al., 2008; Hao et al., 2014; Iinuma et al., 2007) and in forest region affected by urban air mass (Hao et al., 2014). This result thus confirms the major contribution of organic nitrates in SOA formation and also the importance of $NO_3$

chemistry in this process.

**4.3 Products at molecular scale and mechanisms**

In order to provide mechanisms and to propose explanations for the different SOA yields between α- and γ-terpinene, an identification of gas phase products at the molecular scale has been performed by PTR-ToF-MS. The combination of two different ionization modes, with $H_3O^+$ and $NO^+$, for the detection of products allowed an

accurate identification of the molecules. Detected signals in both ionization modes and corresponding raw formula are summarized in Table 4. Products with molecular weights of 114, 152 and 168 g mol$^{-1}$ for γ-terpinene and 58, 152, 168 and 171 g mol$^{-1}$ for α-terpinene have been detected with high intensities. For γ-terpinene the molecular weights of 184, 215 and 229 have also been detected with lower intensities. Many of the detected products are

nitrogenous species which is coherent with high production yields of organic nitrates. To explain these observations, mechanisms have been proposed in Fig. 7 for γ-terpinene and in Fig. 8 α-terpinene. All detected products are framed. Time profiles of PTR-ToF-MS signals were also used to determine whether the products are primary or secondary ones. Typical PTR-ToF-MS profiles are shown in the Fig. S2. First generation products are framed in blue and second generation ones in pink.

**γ-terpinene oxidation scheme**

NO$_3$ radical can react by addition on one of the two double bonds (H-atom abstraction is considered to be negligible), each addition leading to the formation of two possible nitrooxy alkyl radicals. Kerdouci et al., 2014, who developed a structure-activity relationship for VOC+NO$_3$ reactions, suggest that the additions on the two double bonds of γ-terpinene have the same branching ratios. Thus, the four possible nitrooxy alkyl radicals were considered here. However, to facilitate the reading of the mechanism, only two radicals are shown in Fig. 7, considering that in most cases, products obtained are isomers and cannot be distinguished by the analytical techniques used in this work. The mechanism of the two other alkyl radicals is presented in Fig. S4. Nitrooxy alkyl radicals react with O$_2$ to form a peroxy radical (RO$_2$) (reaction 2). However, the formation of an epoxide (reaction 3) with a molecular weight of 152 g mol$^{-1}$ has also been detected both in NO$^+$ (m/z 152) and H$_3$O$^+$ (m/z 153) modes. RO$_2$ can then evolve following different pathways: they can react with another RO$_2$ radical to form a hydroxynitrate and a ketonitrate (reaction 4). The hydroxynitrate (M = 215 g mol$^{-1}$) has been detected at m/z 216 and m/z 214 in H$_3$O$^+$ and NO$^+$ ionization modes respectively. This product is characteristic of the RO$_2$ + RO$_2$ pathway. The ketonitrate (M = 213 g mol$^{-1}$) has been detected at m/z 214 in H$_3$O$^+$ mode and at m/z 243 in NO$^+$ mode (M+30). It should be noticed that this pathway involves an H-atom transfer and so is not possible for tertiary peroxy radicals. RO$_2$ radicals can also react with another RO$_2$ or with NO$_3$ to form alkoxy radicals (RO) (reactions 5 and 5'). Evolution pathways of RO radicals are described by reactions (6), (7) and (8): (i) alkoxy radicals can react with O$_2$ to form a ketonitrate with M = 213 g mol$^{-1}$ (reaction 6), which can also be produced by the pathway 4 (RO$_2$ + RO$_2$). This ketonitrate is thus not characteristic of a single pathway, contrary to the hydroxynitrate. (ii) they can also decompose by a scission of the C(ONO$_2$)-CH(O$^\bullet$) bond (reaction 7, framed in orange) leading to the ring opening and to the formation of a dicarbonyl product with M = 168 g mol$^{-1}$ (detected at m/z 169 in H$_3$O$^+$ mode and at m/z 168 in NO$^+$ mode). (iii) finally, they can decompose by ring opening on the other side of the alkoxy group (reaction 8) leading to the formation of an alkyl radical which reacts to form a trifunctional compound (two carbonyl and one nitrate groups) of molecular weight MW = 229 g mol$^{-1}$ (framed in green in Fig. 7). This product has been detected with a weak signal at m/z 230 in H$_3$O$^+$ ionization mode but was not observed in NO$^+$ mode. Because quantification of these products was not possible, branching ratios between these different pathways could not be determined.

The SAR developed by Vereecken and Peeters, 2009 which is based on DFT calculations, has been used to estimate the energy barriers of the various reaction pathways of the alkoxy radicals. Energy barriers for reactions 7 and 8 appear to be similar (E$_{b,7}$ = 6.0 kcal mol$^{-1}$ et E$_{b,8}$ = 6.5 kcal mol$^{-1}$, with an error estimated by the authors of 0.5 kcal mol$^{-1}$), leading to similar branching ratios for the two possible ring openings. The loss of methyl or isopropyl group presents an energy barrier significantly higher (10.2 kcal mol$^{-1}$) whatever the alkoxy radical considered. These pathways appear minor compared to the ring opening and this is in good agreement with the fact that acetone was not detected during the experiments. It should also be noticed that peroxynitrates (RO$_2$NO$_2$), which have a

characteristic absorption in the IR region, were not detected in our experiments, neither in the gaseous phase, nor in the aerosol one. This suggests that that $RO_2 + NO_2$ reactions are minor pathways.

Primary products that still have a double bond can also react with nitrate radical to form second generation products, framed in pink in the mechanism. This is confirmed by time profiles of primary products MW = 152 and 168 g mol$^{-1}$ which are decreasing with time (see Fig. S2) and by the detection of secondary products with MW = 114, 86, 184, 247 and 245 g mol$^{-1}$. The product with MW = 184 g mol$^{-1}$ correspond to a second generation epoxide. Other products can be explained by the reaction of di-carbonyl compounds with $NO_3$ radical. The compounds with

MW = 114 g mol$^{-1}$ and 86 g mol$^{-1}$ correspond to second generation di-carbonyl products formed by the decomposition of RO radicals (reaction 7-2). Finally the $RO_2 + RO_2$ reaction (reaction 4-2) and the $RO + O_2$ reaction (reaction 6-2) can lead to the formation of highly functionalized products with MW = 247 g mol$^{-1}$ and MW = 245 g mol$^{-1}$.

Vapor pressures and percentage of partitioning into the aerosol phase were calculated as described in section 2.4

for aerosol mass loading of 800 µg.m$^{-3}$, typical of an experiment end, and are also presented next to the molecules in the Fig. 7. Among the first generation products, two types of products have low vapor pressures and can thus participate to SOA formation: first, the hydroxynitrate (e.g. MW = 215 g mol$^{-1}$) which is characteristic of the $RO_2 + RO_2$ pathway (Fig. 7 - pathway 4). It was estimated to partition at 40% in particle phase. Secondly, trifunctional molecules (Fig. 7 - pathway 8) are expected to have very low volatility and to be present mainly in

the aerosol phase (close to 100%). For these two products, the associate partitioning coefficient, $K_p$ following Eq. (10) has been calculated. Considering the uncertainty on $\xi_{aer}^i$ due to the vapor pressure estimation, they can vary from $5.7 \times 10^{-4}$ to $2.0 \times 10^{-3}$ m$^3$ µg$^{-1}$ for the hydroxynitrate (MW = 215 g mol$^{-1}$) and from $5.3 \times 10^{-3}$ to $2.1 \times 10^{-2}$ m$^3$ µg$^{-1}$ for the trifunctional compound (MW = 229 g mol$^{-1}$). They appear to be consistent with the partitioning coefficients found with the two product model from Eq. (8) ( $K_{p,1} = 3.4\ 10^{-3}$ m$^3$ µg$^{-1}$ and $K_{p,2} = 4.5 \times 10^{-2}$ m$^3$ µg$^{-1}$

), especially by considering the associated uncertainty estimated on $K_p$. One can then consider that the two groups of products used by the two products model can be constituted by the hydroxynitrate and the trifunctional compounds, or by similar products.

Other first generation products are estimated to play a minor role in SOA formation. For secondary products, multifunctional products (with 4 chemical groups) are estimated to be between 80 and 100 % in the particle phase.

Other secondary products which are formed by fragmentation processes (dicarbonyl compounds) are expected to be volatile.

To conclude, the oxidation of γ-terpinene by $NO_3$ radical leads to the formation of several functionalized products and particularly to multifunctional organic nitrates which were detected in both phases. Most detected products can be formed by different pathways thus no preferential pathway could clearly be identified. Nevertheless,

products with up to 4 chemical groups (nitrate, carbonyl and alcohol) have been identified and explain the high SOA formation. In particular, the reaction $RO_2 + RO_2 \rightarrow ROH + R(O)$ seems to play a significant role in the SOA formation as hydroxynitrates formed have low vapor pressures (both first and second generation products) with large probability of partitioning into the particle phase.

Results obtained in this study have been compared to those furnished by Slade et al., 2017 who used the CIMS

technique for the detection of products. Mechanisms appear to be very similar: hydroxynitrates, ketonitrates and

dicarbonyl compounds (coming from ring opening) have been observed. No quantification of products has been provided except for hydroxynitrates with an estimated yield in the gas phase of 4 % by using a standard derived from α-pinene. This low yield can be compared with the low total organic nitrate yield found in the same experiments (10 %). In this study, hydroxynitrates were thus found to represent 40 % of total organic nitrates. While detected in our experiments, epoxides have not been detected in this previous study. In addition, Slade et al. 2017 have detected hydroperoxydes formed by $RO_2 + HO_2$ reaction which were not observed in our study. Two hypotheses can explain this difference: (i) in our experiments $HO_2$ concentrations are too small for making this reaction significant compared to those with $RO_2 + RO_2$ or $RO_2 + NO_3$, or (ii) if formed, hydroperoxydes decompose on the stainless steel walls of CESAM chamber. The second hypothesis is expected to be more likely as the loss of hydroperoxydes on CESAM stainless steel walls (particularly $H_2O_2$) has already been observed.

**α-terpinene oxidation scheme**

As described for γ-terpinene, addition of $NO_3$ radical on the double bonds of α-terpinene can lead to the formation of four nitrooxy alkyl radicals. Nevertheless, in this case, the conjugation of the two double bonds allows a delocalization of the single electron and hence, a stabilization of the two corresponding nitrooxy alkyl radicals (see Fig. S3 in SI). Thus, nitrooxy alkyl radicals expected to be the most favorable are those which can undergo an electron delocalization. Note that this delocalization leads to the formation of two tertiary nitrooxy alkyl radicals. Mechanism has been established by considering all possible radicals but for clarity in Fig. 8, only two radicals are presented. The mechanism of the other two alkyl radicals is presented in Fig. S5.

Like for γ-terpinene, nitrooxy alkyl radicals can lose the $NO_2$ group to form epoxides (reaction 3) which were detected by PTR-ToF-MS (m/z 153 in $H_3O^+$ mode and m/z 152 in $NO^+$ mode). They also react with $O_2$ to form $RO_2$ radicals which then evolve through $RO_2 + NO_3$ reaction (reaction 5) and/or through $RO_2 + RO_2$ reaction (reaction 5') to form alkoxy radicals. Here, the formation of the hydroxynitrate (MW = 215 g mol$^{-1}$) was not observed suggesting that the reaction $RO_2 + RO_2 \rightarrow ROH + R(O)$ does not occur. This can be explained by the fact that most favored radicals are tertiary ones and cannot undergo H-shift. From product identification we propose the following decomposition pathways for RO radicals: i) alkoxy radicals can lose the isopropyl group leading to the formation of a cyclic ketonitrate and an isopropyl radical which then evolve to form acetone. The cyclic ketonitrate and the acetone have been detected by PTR-ToF-MS with high intensity signals in both ionization modes. Acetone was also detected by FTIR but its concentration was close to the detection limit of 10 ppb and formation yield could not be precisely measured. Nevertheless, by considering the 10 ppb detection limit, the acetone formation yield is expected to be below 3 %; ii) they can also decompose by a scission of the C(ONO₂)-CH(O•) bond (reaction 7, framed in orange) leading to a ring opening and to the formation of a dicarbonyl product with MW = 168 g mol$^{-1}$. This compound was detected with high signals by PTR-ToF-MS in $H_3O^+$ and $NO^+$ modes, respectively at m/z 169 and m/z 168; iii) the formation of trifunctional species (one nitrate group and two carbonyl group) with MW = 229 g mol$^{-1}$, coming from the ring opening on the other side of alkoxy group (reaction 8) has also been observed but it needs confirmation. Indeed, mass m/z 230 has been detected in $H_3O^+$ mode, but neither m/z 229 nor 259 were detected in $NO^+$ mode. As for γ-terpinene, they can be formed with low concentrations and/or be mostly in particle phase leading to weak signals with PTR-ToF-MS.

The SAR proposed by Vereecken and Peeters, 2009 allowed estimating the energy barriers of the different evolution pathways of alkoxy radicals. It suggests that ring openings (reactions 7 and 8) are the two most likely pathways with similar branching ratios (energy barriers are respectively 6.0 and 6.5 kcal mol$^{-1}$). As for γ-terpinene, reaction 7' corresponding to the loss of the isopropyl group appears to be less favorable with an energy barrier of 10.2 kcal mol$^{-1}$. This is in agreement with the low acetone formation yield. This is also in agreement with the detection of the trifunctional species for which the weak signal observed can be explained by its low volatility. This product is the only primary product expected to contribute to SOA formation (with $\xi_{aer}^{i} = 50$ %). As a reminder, the associated partitioning coefficient $K_p$ for this trifunctional compound (MW = 229 g mol$^{-1}$) was estimated to range between $5.3 \times 10^{-3}$ and $2.1 \times 10^{-2}$ m$^3$ μg$^{-1}$. It can correspond to the $K_{p,2}$ estimated by the two-product model in Eq. (8) $\left(K_{p,2} = 3.5 \times 10^{-2}\ m^3\ \mu g^{-1}\right)$. The second partitioning coefficient in the two-product model ($K_{p,1} = 4.5 \times 10^{-1} m^3\ \mu g^{-1}$) cannot be attributed, and it can be explained by i) the fact that this product can be totally in the particle phase and thus not detectable in the gas phase and ii) the very low production of SOA, leading to a low precision on the fit.

The only secondary products identified are epoxides that can come from the reaction of the primary epoxide with NO$_3$ radical, but also from the oxidation of the unsaturated dicarbonyl compounds. In the last case, they are epoxides coming from the loss of -NO$_2$ from alkyl radicals (reaction 3-2). Other secondary products were expected, but not detected. Signal m/z 243 has been detected in NO$^+$ mode but this signal could not be attributed to a product.

In conclusion, products detected with highest signals are cyclic ketonitrates, dicarbonyl compounds and epoxides. These three families of products have high vapor pressures and are thus not expected to significantly contribute to SOA formation. Trifunctional products with low vapor pressures have also been detected with low signals, suggesting low formation yields or strong partition in aerosol phase. Low SOA yields measured for α-terpinene suggest that the first explanation is more favorable.

**5 Comparative discussion**

Even though α- and γ-terpinene have similar chemical structure only differing by the position of the double bonds, their reactions with nitrate radical have different consequences especially regarding SOA formation. Mean SOA and organic nitrate yields obtained for both compounds are presented in Table 5. Several previous studies on BVOC + NO$_3$ reactions suggest a correlation between organic nitrate yield and SOA formation (Fry et al., 2014; Hallquist et al., 1999). α-pinene presents indeed a low organic nitrate yield, in good agreement with a SOA yield close to zero, when limonene and Δ-carene both present high SOA and organic nitrate yields. In our study, Y$_{ONg}$ and Y$_{ONp}$ for α- and γ-terpinene are similar regarding the uncertainty. Thus α-terpinene does not follow this correlation, as it produces a high amount of organic nitrates, but almost no SOA.

To interpret the difference in SOA formation, the mechanisms have to be compared: for γ-terpinene, the major SOA production can be explained by the formation of (i) a primary hydroxynitrate coming from RO$_2$ + RO$_2$ pathway and (ii) a primary trifunctional nitrate formed by the decomposition of alkoxy radical and (iii) detection of secondary products, all presenting very low vapor pressures. On the other hand, for α-terpinene (i) no hydroxynitrate and (ii) no secondary products with low vapor pressure were detected in the oxidation products. The absence of hydroxynitrate may be explained by the fact that, due to relocation of the free electron by

mesomeric effect for alkyl radicals, the most stable radicals are tertiary and cannot undergo H-shift to produce the hydroxynitrate. Hence, two aspects of the mechanism appear to be critical for the SOA formation: i) the peroxy radical reaction pathways: when the carbon that bears the radical group, has a hydrogen available, the reaction $RO_2 + RO_2 \rightarrow ROH + R(O)$ can occur, leading here to the formation of an hydroxynitrate. Due to hydrogen bonds, this product has very low vapor pressure. This has already been reported by Ng et al. 2008 who compared isoprene SOA formation for reactions $RO_2 + RO_2$ et $RO_2 + NO_3$. The study reported higher formation of SOA for the pathway $RO_2 + RO_2$ due to the formation of the specific hydroxynitrate and its secondary reaction. In our study, this product was detected for γ-terpinene, but not for α-terpinene for which we expect that most stable radicals are tertiary ones and cannot undergo this reaction pathway. ii) the alkoxy radical reaction pathways: several decomposition pathways can occur, leading to products which have very different volatilities. If they decompose by a scission of the $C(ONO_2)$-$CH(O^\bullet)$ bond, dicarbonyl products which are volatile, are formed. If the decomposition occurs by a ring opening on the other side of the alkoxy group, keto-nitrooxy-alkyl radicals are formed, which then evolve towards the formation of low vapor trifunctional species. This point has already been raised by Kurten et al., 2017 who performed computational calculations on the alkoxy reaction pathways to explain the low SOA yield observed for α-pinene, in comparison to Δ-carene. The authors suggest that for Δ-carene, the decomposition of alkoxy radicals can lead to the formation of keto-nitrooxy-alkyl radicals, whereas for α-pinene, the alkoxy radicals decompose almost exclusively to form the dicarbonyl compound.

The study of Claflin and Ziemann, 2018 showed that the hydroxynitrates formed by the $RO_2 + RO_2$ pathway and the carbonyl compounds, via an acid-catalyzed particle phase reaction lead to the formation of acetal dimers and trimers. No molecular analysis of the particle phase, except for the organic nitrates we conducted. If polymers are formed in the particle phase, for example acetal dimers and trimers, which have a nitrate group, they cannot be distinguished from the monomers. For γ-terpinene, hydroxynitrates and carbonyl nitrates were detected in the gas phase, which have low enough volatility to go in particle phase and contribute the most to SOA formation. They can then react to form acetal dimers or trimers in the particle phase, but with no possible detection. For α-terpinene, no hydroxynitrate formation was detected; the formation of these dimers is expected to be negligible. This is in good agreement with the low SOA yields for this compound.

This study also showed the importance of $RO_2 + RO_2$ reaction and alkoxy decomposition, which are the key point in α- and γ-terpinene chemistry. The importance of isomerization and acid-catalyzed particle phase reaction has not been proved but is coherent with the results.

It is also interesting to compare the reactivity of γ-terpinene and α-terpinene to those of other monoterpenes. γ-terpinene produces large amounts of SOA (with yields ranging between 20 and 40 %) similarly to limonene, β-pinene, Δ-3-carene and sabinene (Fry et al., 2009, 2014; Griffin et al., 1999; Hallquist et al., 1999; Moldanova and Ljungstrom, 2000; Spittler et al., 2006). On the other hand, α-terpinene produces small amounts of SOA with yields around 1 %. It can be compared to α-pinene, with yields between 0 and 16 % (Hallquist et al., 1999; Spittler et al., 2006, Nah et al. 2016, Fry et al. 2014, Perraud et al. 2010). An explanation for this low SOA yield regarding α-pinene has been given by Kurten et al., 2017.

The two studied compounds have organic nitrate yields around 50 % which appear similar to those measured for other BVOCs, such as limonene, with yields between 30 and 72 % (Hallquist et al., 1999; Spittler et al., 2006) or

β-pinene, between 22 and 74% (Boyd et al., 2015; Fry et al., 2014; Hallquist et al., 1999). Within the uncertainties, they also appear similar to those obtained for Δ-carene (68-77%, Fry et al., 2014; Hallquist et al., 1999) and isoprene (62-78 %, Rollins et al., 2009). For most BVOCs, NO₃ radical chemistry is a major organic nitrate precursor. Only α-pinene has been shown to produce less organic nitrates, between 10 and 25 % (Berndt and Boge, 1997; Fry et al., 2014; Hallquist et al., 1999; Spittler et al., 2006; Wangberg et al., 1997).

One interesting fact is that for both compounds, epoxides have been detected, whereas it is usually admitted that their formation is favored only at low oxygen concentration (Berndt and Böge, 1995). Many studies did not detect these compounds (Jaoui et al., 2013; Slade et al., 2017; Spittler et al., 2006), but their formation has already been observed by Skov et al., 1994, which studied NO₃ radical oxidation of several alkenes and isoprene. Wangberg et al., 1997 also showed low epoxides yields (3 %) for α-pinene and Ng et al., 2008 (>1 %) for isoprene. In our study, epoxides were not quantified, but based on previous studies, their formation yields are expected to be low.

## 6 Conclusions & atmospheric impacts

In summary, this work provides kinetic and mechanistic data on the oxidation by NO₃ radical of α- and γ-terpinene, using simulation chambers. The two compounds present very similar chemical structures (the same carbon skeleton and two double bonds which are conjugated in the case of α-terpinene, and not for γ-terpinene) and this work aimed at highlighting the influence of the structure on the reactivity, in particular on the SOA formation.

Absolute and relative kinetic determinations have been performed. This study provides the first absolute determination for α-terpinene. Both α- and γ-terpinene appear to be very reactive towards nitrate radical due to the presence of two double bonds. α- terpinene is particularly reactive with NO₃ radicals due to the conjugation of the double bonds while γ- terpinene is four times less reactive.

As far as we know, this study is the first mechanistic study for the oxidation of α-terpinene by NO₃ radicals. Our study has confirmed that the NO₃ oxidation of α- and γ-terpinene produces large amounts of organic nitrates (with overall yields ~ 50%) which have been shown to be present in both gas and aerosol phases. Nevertheless, major differences in the SOA formation for the two compounds have been pointed out despite their similar structure. γ-terpinene has been shown to be an efficient SOA precursor, when α-terpinene is a poorly efficient SOA precursor. To explain these differences, a molecular scale study has been conducted and two reaction pathways have been shown to play a key role in the SOA formation: i) the peroxy radical reaction pathways leading to the formation of low volatility hydroxynitrates, which were detected for γ-terpinene but not for α-terpinene ii) the alkoxy radical scission pathways which can either form high volatility dicarbonyl compounds or low volatility trifunctional products, depending on where the scission occurs.

The atmospheric lifetimes of the two compounds have been estimated by using typical nighttime NO₃ concentration (10 ppt) and low insolation diurnal concentration (0.1 ppt, Corchnoy and Atkinson, 1990). These lifetimes are presented and compared to those estimated for oxidation by OH and ozone in Table 6. Prior to the discussion, it is important to remind that monoterpenes are intensively emitted during both day and night (Lindwall et al., 2015). From Table 6, it can be observed that the two monoterpenes exhibit very short lifetimes towards NO₃ radical for nighttime conditions (40 s for α-terpinene and 2 min for γ-terpinene) confirming that NO₃ oxidation is a major sink for these compounds. As expected, lifetimes estimated for low sunlight diurnal conditions are longer

(few hours), but are still fairly short. By comparison with lifetimes estimated for other oxidants, it is concluded that all three oxidants are very efficient sinks. For low insolation diurnal conditions, even though diurnal chemistry is clearly led by OH and $O_3$, $NO_3$ oxidation is not negligible. This result can be compared to the modeling study of Forkel et al., 2006, which has shown that $NO_3$ oxidation is an important sink of BVOCs even during the day under canopy, with low luminosity and considering a calculated mixing ratio for $NO_3$

of 3 ppt).

The short lifetimes indicate that oxidation products are formed close to the BVOCs emission areas. If all three oxidants are major sinks, the products formed by the different processes are very different. Organic nitrates can also be formed by OH oxidation through $RO_2$ + NO reactions but yields are much lower. Lee et al. 2006 have investigated the products formed by the OH oxidation of 16 terpenoids, including α- and γ-terpinene, in presence

of NOx. Organic nitrate yields were shown to be less than 1%. So a major impact of BVOC oxidation by $NO_3$ radicals is the formation of organic nitrates which are known to act as NOx reservoirs. In our study, dicarbonyl compounds have also been shown to be formed but the same compounds have been detected as major products of the OH chemistry of α- and γ-terpinene by Lee et al. 2006. So, this is not a special feature of the nighttime chemistry.

SOA yields produced by $NO_3$ oxidation can be compared to those formed by ozonolysis and OH oxidation. Friedman and Farmer, 2018, Griffin et al., 1999, Lee et al., 2006 have measured the SOA yields for the OH oxidation of several terpenes, including α- and γ-terpinene. In the experiments performed by Griffin et al., 1999, mixing of terpenes were introduced leading to overall SOA yields which were found to be 4 % at 10 µg m$^{-3}$ and 20 % at 400 µg m$^{-3}$. Lee et al., 2006 have provided final SOA yields for α- and γ-terpinene and shown that they

are below 20 %. Finally, the study of Friedman and Farmer, 2018, was performed with low NOx conditions and SOA yields were found to be very low (≤ 1 %) at $M_0$ = 10 µg m$^{-3}$. Regarding these results, the oxidation by $NO_3$ appears to be a much more efficient SOA source than the OH oxidation. This observation has already been made by several previous studies (Hallquist, Wangberg, and Ljungstrom 1997; Griffin et al. 1999; Spittler et al. 2006; Ng et al. 2008; Fry et al. 2014; Boyd et al. 2015, Slade et al. 2017). Regarding the ozonolysis of α- and γ-terpinene,

there is, to our knowledge, no data on SOA yields in the literature. However, more generally, the ozonolysis of BVOCs is known to be an important source of SOA.

In conclusion, the most important impacts of this chemistry rely on the formation of large amounts of organic nitrates (present in both gas and aerosol phases) and SOA. Organic nitrates play a key role in tropospheric chemistry because they behave as NOx reservoirs, carrying reactive nitrogen in remote areas. Their chemistry in

gas and aerosol phases is nevertheless still not well documented. Considering that our study shows a large production of multifunctional organic nitrates, it is necessary to better understand their reactivity in order to better evaluate their impacts. Formation of SOA seems on the other hand, strongly dependent on the structure of the BVOC. Studies at molecular scale are thus mandatory to better evaluate the impact of this chemistry on the SOA formation.

*Data availability:* Rate constant for the $NO_3$ oxidation of α- and γ-terpinene are available Table 2. It is also available through the Library of Advanced Data Products (LADP) of the EUROCHAMP data center (https://data.eurochamp.org/data-access/ gas phase-rate-constants/, last access: 01 May 2020, Fouqueau et al.,

2020b). Kinetic and mechanistic simulation chamber experiments are available through the Database of Atmospheric Simulation Chamber Studies (DASCS) of the EUROCHAMP data center (https://data.eurochamp.org/data-access/chamber-experiments/, last access: 01 May 2020, Fouqueau et al., 2020c).

*Author contributions:* BPV and MCi coordinated the research project. AF, BPV, MCi and JFD designed the experiments in the simulation chambers. AF performed the experiments with the technical support of MC and EP and performed the data treatment and interpretation with MCi and BPV. AF, BPV and MCi wrote the paper, and AF was responsible for the final version of the paper. All coauthors revised the content of the original manuscript and approved the final version of the paper.

*Competing interests.* The authors declare that they have no conflict of interest.

*Special issue statement.* This article is part of the special issue "Simulation chambers as tools in atmospheric research (AMT/ACP/GMD inter-journal SI)". It is not associated with a conference.

*Acknowledgements.* The authors thank Marie Camredon (LISA, Créteil, France) for helping with the GECKO-A website, Marie-Thérèse and Jean-Claude Rayez (ISM, Bordeaux, France) for helping understanding the reactivity with theoretical calculation.

*Financial support.* This work was supported by the French national programme LEFE/INSU and by the European Commission through the program H2020 Research Infrastructures (EUROCHAMP-2020; grant no. 730997).

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

**Table 1: Experimental conditions of kinetic experiments. [BVOC] is the initial concentration of BVOC, Ref. is the reference compound and [Ref.] is the initial concentration of the reference compound. For [$N_2O_5$] the number of punctual injections is indicated in brackets.**


| BVOC | Date (yyyy/dd/mm) | Method* | [BVOC] (ppb) | Ref. | [Ref.] (ppb) | [$N_2O_5$] (ppb) | [$NO_2$] (ppb) |
|---|---|---|---|---|---|---|---|
| γ-terp. | 2015/27/11 | RR | 610 | 2-methyl-2-butene | 640 | 100 (4 inj.) | - |
| | 2016/05/01 | RR | 680 | 2-methyl-2-butene | 600 | 100; 200 (4 inj.) | - |
| | 2016/06/01 | RR | 1310 | 2-methyl-2-butene | 1290 | 100 (2 inj.); 200 (4 inj.) | - |
| | 2016/07/01 | RR | 1290 | 2,3-dimethyl-2-butene | 1290 | 100; 200 (4 inj.); 300 | - |
| | | RR | 1310 | 2,3-dimethyl-2-butene | 1200 | 200 (5 inj.) | - |
| | 2018/30/01 | AR | 20 | - | - | 20 | 630 |
| | | AR | 35 | - | - | 20 (3 inj.) | 620 |
| | 2018/01/02 | AR | 34 | - | - | 20 (2 inj.) | 470 |
| α-terp. | 2018/29/01 | AR | 14 | - | - | 20 (2 inj.) | 560 |
| | | AR | 41 | - | - | 20; 40 | 480 |
| | 2018/01/02 | AR | 48 | - | - | 20 (3 inj.) | 460 |

* RR: relative rate determination; AR: absolute rate determination

**Table 2 : Rate constants for the NO₃-initiated oxidation of γ-terpinene and α-terpinene: results from this study and comparison with literature.**

| BVOC | k (cm³ molecule⁻¹ s⁻¹) | (k$_{COVB}$/k$_{ref}$) | Study (method) |
|---|---|---|---|
| **γ-terpinene** | **$(3.0 \pm 0.9) \times 10^{-11}$** | | **This study (AR[a])** |
| | **$(3.0 \pm 1.1) \times 10^{-11}$** | **$(0.60 \pm 0.01)$** | **This study (RR[b] : 2,3-dimethyl-2-butene)** |
| | **$(2.7 \pm 0.6) \times 10^{-11}$** | **$(2.8 \pm 0.1)$** | **This study (RR[b] : 2-methyl-2-butene)** |
| | $(2.97 \pm 0.9) \times 10^{-11}$ | $(3.1 \pm 0.1)$ | Atkinson et al. 1985 (RR[b] : 2-methyl-2-butene) |
| | $(2.4 \pm 0.7) \times 10^{-11}$ | | Martinez et al. 1999 (AR[a]) |
| | $2.2 \times 10^{-11}$ | | Estimated with SAR (Kerdouci et al., 2014) |
| **α-terpinene** | **$(1.2 \pm 0.3) \times 10^{-10}$** | | **This study (AR[a])** |
| | $(1.6 \pm 0.6) \times 10^{-10}$ | $(3.2 \pm 0.1)$ | Atkinson et al. 1985 (RR[b] : 2,3-dimethyl-2-butene) |
| | $(0.9 \pm 0.4) \times 10^{-10}$ | $(1.8 \pm 0.1)$ | Berndt et al. 1996 (RR[b] : 2,3-dimethyl-2-butene) |
| | $1.0 \times 10^{-10}$ | | Estimated with SAR (Kerdouci et al., 2014) |

[a]: **Absolute rate determination;** [b] : **Relative rate determination**

**Table 3: Experimental conditions, ONs and SOA yields for mechanistic experiments conducted in CESAM chamber. The use of an instrument is shown by a cross, the non-use by a dash.**

| BVOC | Date (yyyy/mm/dd) | [BVOC]$_i$ (ppb)* | N$_2$O$_5$ injection (concentration and/or duration) | PTR-ToF-MS (NO$^+$) | PTR-ToF-MS (H$_3$O$^+$) | Filter sampling & analysis | $Y_{ONg,}$ molar | $Y_{ONp,}$ molar | $Y_{ON(g+p),}$ molar | $Y_{ONp,}$ mass | $Y_{SOA}$ mass | $Y_{ONp, mass}$ / $Y_{SOA,mass}$ |
|---|---|---|---|---|---|---|---|---|---|---|---|---|
| γ-terp. | 2016/02/16 | 430 | Stepwise (100 ppb, 3 inj., 4 min) | x | - | - | 0.33 ± 0.03 | - | - | - | 0.41 ± 0.12 | - |
| | 2016/02/17 | 230 | Stepwise (800 ppb, 3 inj., 3min) | x | - | - | 0.57 ± 0.06 | - | - | - | 0.43 ± 0.13 | - |
| | 2016/02/18 | 130 | Continuous (21 min) | x | - | - | 0.46 ± 0.05 | - | - | - | 0.39 ± 0.12 | - |
| | 2016/02/19 | 150 | Continuous (73 min) | x | - | - | 0.33 ± 0.04 | - | - | - | 0.22 ± 0.07 | - |
| | 2017/03/21 | 380 | Continuous (15 min) | x | - | - | 0.41 ± 0.04 | - | - | - | - | - |
| | 2017/03/23 | 250 | Continuous (40 min) | - | - | - | 0.33 ± 0.03 | - | - | - | 0.32 ± 0.11 | - |
| | 2017/03/24 | 340 | Stepwise (400 ppb, 1 inj., 4 min) | - | x | x | 0.65 ± 0.07 | 0.08 ± 0.03 | 0.73 ± 0.35 | 0.13 ± 0.05 | 0.42 ± 0.12 | 0.31 ± 0.19 |
| | 2017/04/25 | 490 | Continuous (36 min) | - | x | x | 0.48 ± 0.06 | 0.07 ± 0.03 | 0.55 ± 0.31 | 0.11 ± 0.04 | 0.22 ± 0.06 | 0.50 ± 0.30 |
| | 2017/04/26 | 410 | Continuous (42 min) | - | x | x | 0.59 ± 0.08 | 0.06 ± 0.02 | 0.65 ± 0.31 | 0.10 ± 0.04 | 0.25 ± 0.07 | 0.40 ± 0.24 |
| | 2017/12/13 | 70 | Continuous (31 min) | x | x | x | 0.35 ± 0.03 | 0.011 ± 0.004 | 0.36 ± 0.16 | 0.02 ± 0.01 | 0.13 ± 0.04 | 0.23 ± 0.14 |
| | 2017/12/20 | 130 | Continuous (61 min) | x | x | x | 0.44 ± 0.04 | 0.020 ± 0.008 | 0.46 ± 0.23 | 0.03 ± 0.01 | 0.29 ± 0.09 | 0.07 ± 0.04 |
| α-terp. | 2017/03/27 | 310 | Continuous (20 min) | - | x | - | 0.38 ± 0.12 | - | - | - | 0.02 ± 0.01 | - |
| | 2017/03/28 | 350 | Continuous (62 min) | - | x | x | 0.23 ± 0.07 | 0.04 ± 0.01 | 0.06 ± 0.02 | 0.06 ± 0.02 | 0.07 ± 0.02 | 0.86 ± 0.51 |
| | 2017/03/29 | 340 | Continuous (64 min) | x | - | x | 0.44 ± 0.06 | 0.03 ± 0.01 | 0.05 ± 0.02 | 0.05 ± 0.02 | 0.04 ± 0.01 | 1.25 ± 0.75 |
| | 2017/04/19 | 360 | Continuous (48 min) | x | - | - | 0.46 ± 0.04 | - | - | - | 0.006 ± 0.002 | - |
| | 2017/04/20 | 380 | Continuous (31 min) | x | - | x | 0.46 ± 0.05 | 0.03 ± 0.01 | 0.05 ± 0.02 | 0.05 ± 0.02 | 0.05 ± 0.01 | 1.0 ± 0.6 |
| | 2017/12/14 | 61 | Continuous (20 min) | x | x | - | 0.3 ± 0.11 | - | - | - | - | - |
| | | 110 | Continuous (23 min) | x | x | - | 0.23 ± 0.06 | - | - | - | 0.05 ± 0.01 | - |
| | 2017/12/19 | 160 | Continuous (40 min) | x | x | x | 0.25 ± 0.04 | 0.012 ± 0.006 | 0.03 ± 0.02 | 0.03 ± 0.02 | 0.04 ± 0.01 | 1.25 ± 0.75 |

*for all experiments, the BVOC was totally consumed.

**Table 4: Products detected for γ-terpinene (top table) and α-terpinene (bottom table) with PTR-ToF-MS $H_3O^+$ and $NO^+$ ionization modes: formula and molar masses, detected masses, ionization processes ($H^+$ : proton adduct, $NO^+$ : $NO^+$ adduct, CT: charge transfer and PL : proton loss), peak intensity, and comportments.**

| Molecule | | $H_3O^+$ ionization mode | | | | $NO^+$ ionization mode | | | |
|---|---|---|---|---|---|---|---|---|---|
| Raw formula | M (g/mol) | m/z | Process | Intensity | Behavior | m/z | Process | Intensity | Behavior |
| $C_4H_6O_2$ | 86 | 87.036 | $H^+$ | + | Secondary | 86.0182 | CT | + | Secondary |
| $C_6H_{10}O_2$ | 114 | 115.0565 | $H^+$ | +++ | Prim.+Sec. | 114.0568 | CT | +++ | Secondary |
| $C_{10}H_{16}O$ | 152 | 153.1062 | $H^+$ | +++ | Primary | 152.0994 | CT | ++ | Primary |
| $C_{10}H_{16}O_2$ | 168 | 169.0931 | $H^+$ | +++ | Primary | 168.0859 | CT | ++ | Primary |
| $C_{10}H_{16}O_3$ | 184 | 185.1034 | $H^+$ | ++ | Secondary | 184.1085 | CT | + | Secondary |
| $C_{10}H_{14}NO_4$ | 213 | 214.1006 | $H^+$ | + | Primary | 243.1696 | $NO^+$ | + | Primary |
| $C_{10}H_{16}NO_4$ | 215 | 216.0546 | $H^+$ | ++ | Primary | 214.0814 | PL | + | Primary |
| $C_{10}H_{15}NO_5$ | 229 | 230.1013 | $H^+$ | ++ | Primary | / | / | / | / |
| $C_{10}H_{14}NO_6$ | 245 | 246.095 | $H^+$ | + | Secondary | / | / | / | / |
| $C_{10}H_{16}NO_6$ | 247 | 248.0925 | $H^+$ | + | Secondary | 246.1412 | PL | + | Primary |

| Molecule | | $H_3O^+$ ionization mode | | | | $NO^+$ ionization mode | | | |
|---|---|---|---|---|---|---|---|---|---|
| Raw formula | M (g/mol) | m/z | Process | Intensity | Behavior | m/z | Process | Intensity | Behavior |
| $C_3H_6O$ | 58 | 59.04574 | $H^+$ | +++ | Primary | 88.0411 | $NO^+$ | ++ | Primary |
| $C_{10}H_{16}O$ | 152 | 153.0855 | $H^+$ | ++ | Primary | 152.1168 | CT | + | Primary |
| $C_{10}H_{16}O_2$ | 168 | 169.1045 | $H^+$ | ++ | Primary | 168.1197 | CT | +++ | Primary |
| $C_7H_9NO_4$ | 171 | 172.0239 | $H^+$ | ++ | Primary | 171.0759 | CT | ++ | Primary |
| $C_9H_{14}NO_3$ | 184 | 185.1105 | $H^+$ | + | Secondary | 184.1277 | CT | + | Secondary |
| $C_{10}H_{15}NO_4$ | 195 | 196.1193 | $H^+$ | + | Primary | 195.0942 | CT | + | Primary |
| $C_{10}H_{14}O_5$ | 214 | / | / | / | / | 214.1016 | | + | Primary |
| $C_{10}H_{15}NO_5$ | 229 | 230.1288 | $H^+$ | + | Detected | / | / | / | / |
| $C_{10}H_{13}NO_6$ | 243 | / | / | / | / | 243.0858 | $NO^+$ | + | Secondary |

**Table 5: Mean SOA and organic nitrate yields obtained in this study for γ-terpinene and α-terpinene.**

| Compound | $Y_{SOA}$, $(10 \ \mu g \ m^{-3})$ | $Y_{ONg}$ | $Y_{ONp, max}$ | $Y_{ON, total}$ |
|---|---|---|---|---|
| γ-terpinene | 10 % | $47 \pm 10\%$ | $8 \pm 3\%$ | $55 \pm 15\%$ |
| α-terpinene | 1.2 % | $44 \pm 10\%$ | $4 \pm 1\%$ | $48 \pm 12\%$ |

**Table 6: Atmospheric lifetimes of α- and γ-terpinene with respect to their oxidation by NO₃ and OH radicals and by ozone.**

| Compound | $\tau_{NO3}$ * (min) | $\tau_{NO3}$** (min) | $\tau_{OH}$*** (min) | $\tau_{O3}$*** (min) |
|---|---|---|---|---|
| α-terpinene | 0.6 | 57 | 23[1] | 1,2[1] |
| γ-terpinene | 2.3 | 239 | 48[2] | 158[1] |

* calculated with $[NO_3] = 2.5 \times 10^8$ molecule cm$^{-3}$ (10 ppt)

** calculated with $[NO_3] = 2.5 \times 10^6$ molecule cm$^{-3}$ (0.1 ppt)

***calculated with $[OH] = 2 \times 10^6$ molecule cm$^{-3}$ and $[O_3] = 7 \times 10^{11}$ molecule cm$^{-3}$

[1] calculated with rate constant recommended by IUPAC

[2] calculated with rate constant from Atkinson et al., 1986

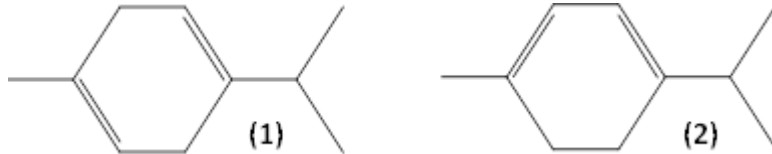

**Figure 1 : Molecular representation of γ-terpinene (1) and α-terpinene (2)**

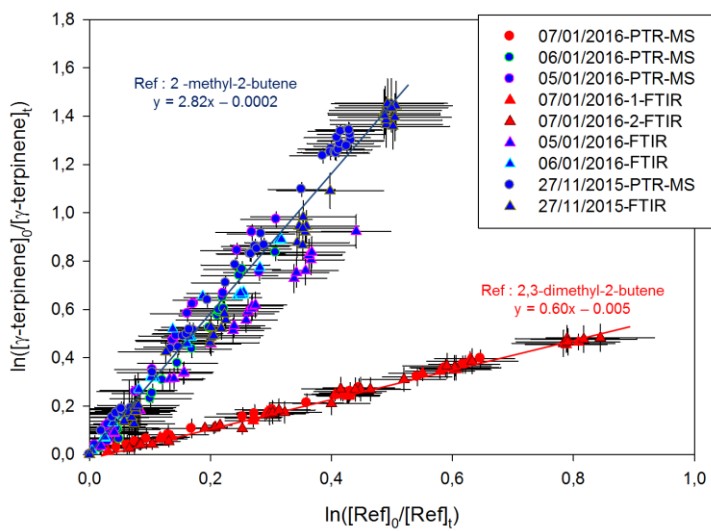

**Figure 2: Relative kinetic plots measured by FTIR (triangle marks) and PTR-ToF-MS (round marks), with 2-methyl-2-butene (blue) and 2,3-dimethyl-2-butene (red) as reference compounds.**


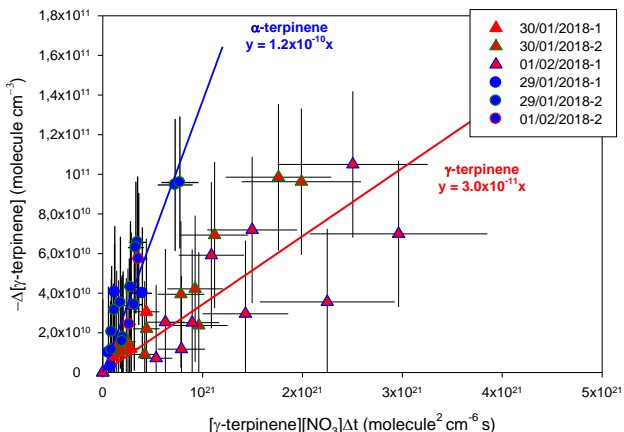

**Figure 3 : Absolute kinetic plots for γ-terpinene (triangle marks) and for α-terpinene (round marks).**

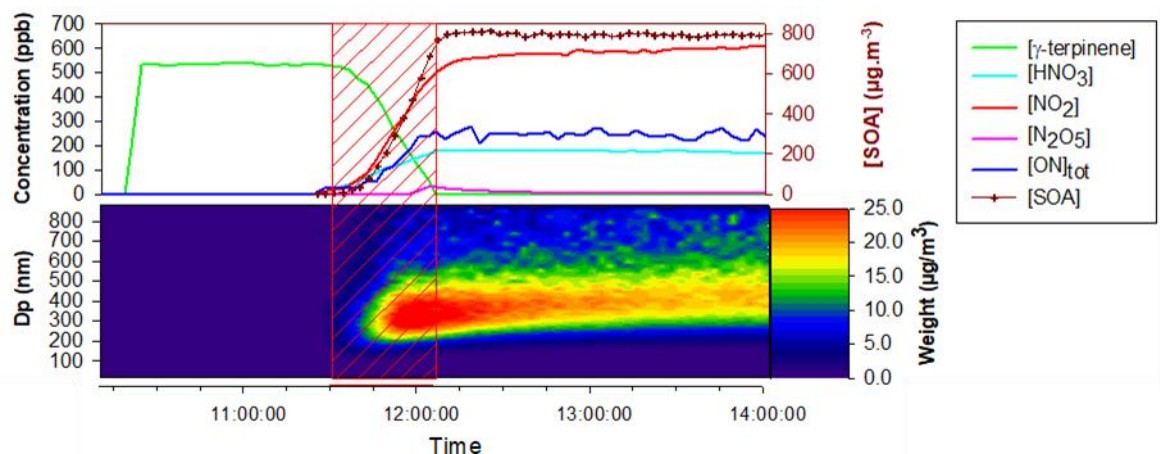


**Figure 4: Time-dependent concentration of gaseous species, aerosol mass (corrected from dilution), SOA size distribution and PTR-ToF-MS signals during a typical experiment of NO₃-initiated oxidation of γ-terpinene (25/04/2017). Red dashed area corresponds to N₂O₅ injection period. Top figure: γ-terpinene, N₂O₅, NO₂, HNO₃ and total ONs from FTIR and SOA mass concentration from SMPS; bottom figure : SOA size distribution in mass concentration from SMPS.**


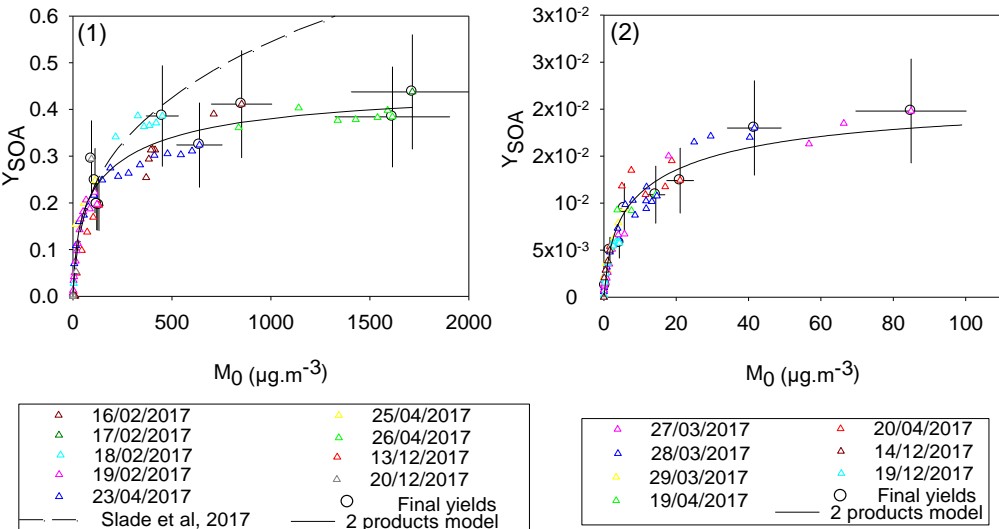

**Figure 5: SOA yield as a function of the organic aerosol mass concentration measured for γ-terpinene (left graph) and for α-terpinene (right graph). Final yields (circle marks) are shown with uncertainties. Data were fitted with a two-product model (black curve) and compared with the study of Slade et al., 2017 (dashed curve).**

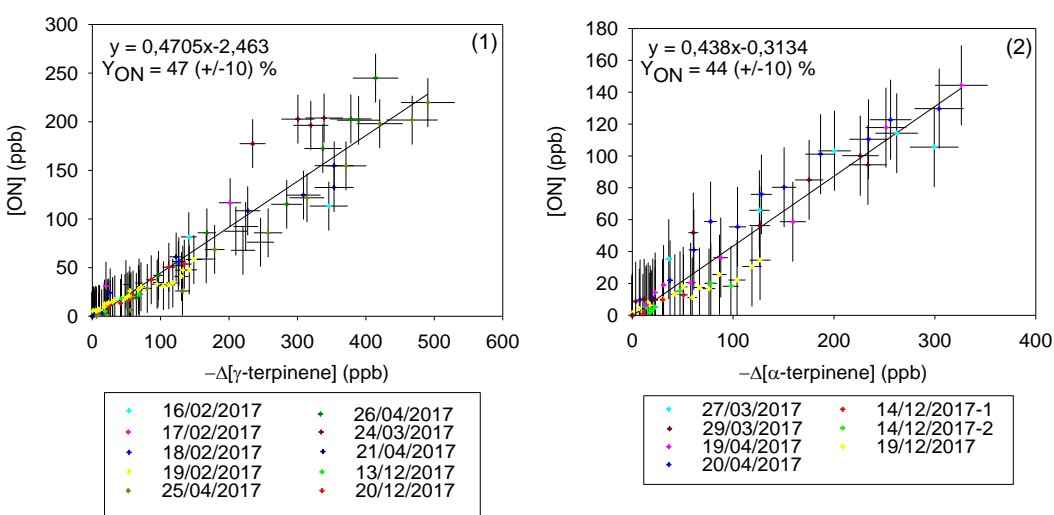


**Figure 6: Gas phase organic nitrates production *vs* loss of γ-terpinene (1) and α-terpinene (2).**





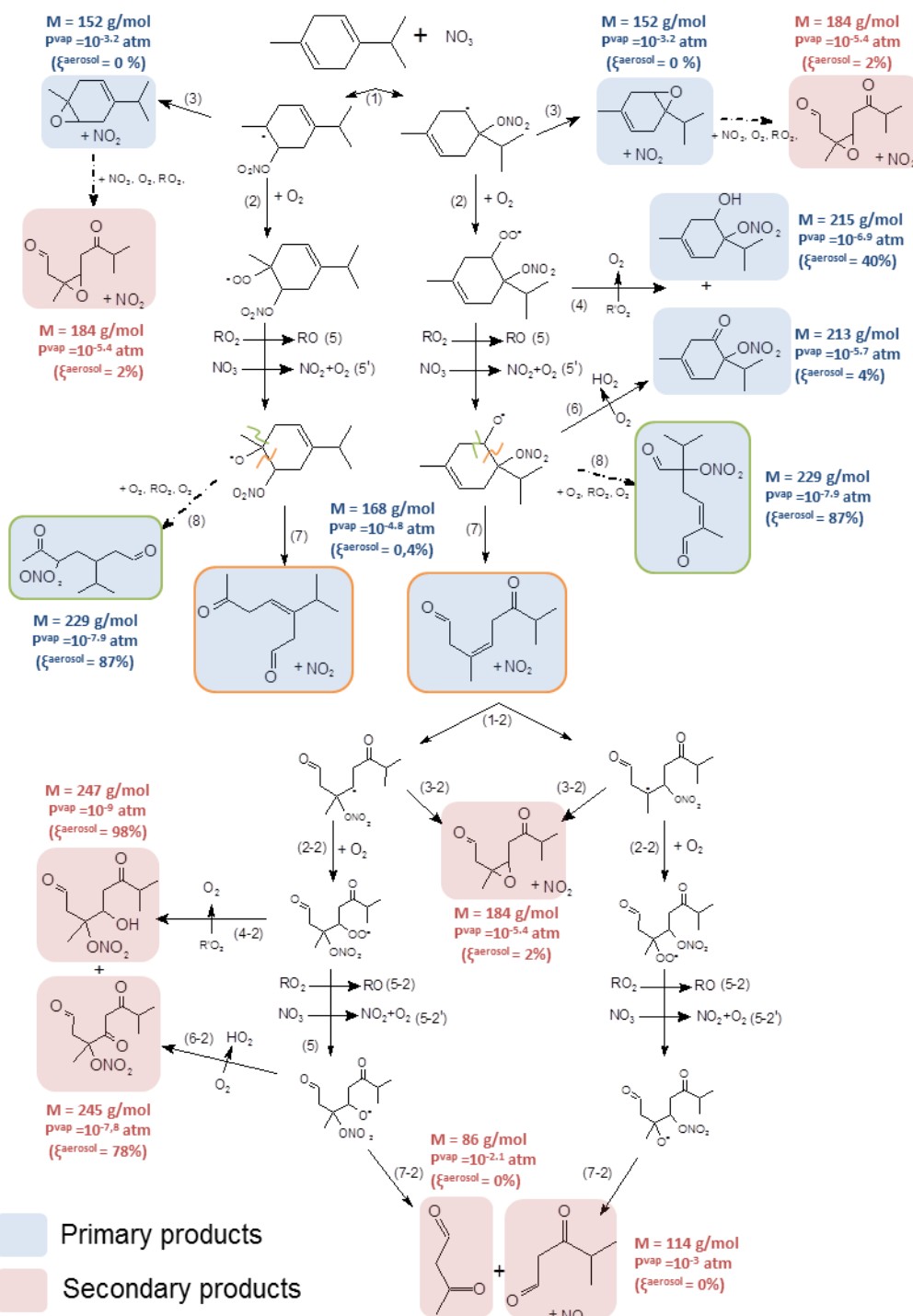

**Figure 7: Proposed mechanism for γ-terpinene. First generation products are squared in blue and second generation ones in red. Alkoxy fragmentation products are squared according to the location of the fragmentation. Molecular weight, vapor pressures and the gas/particle partition are shown next to the molecules.**

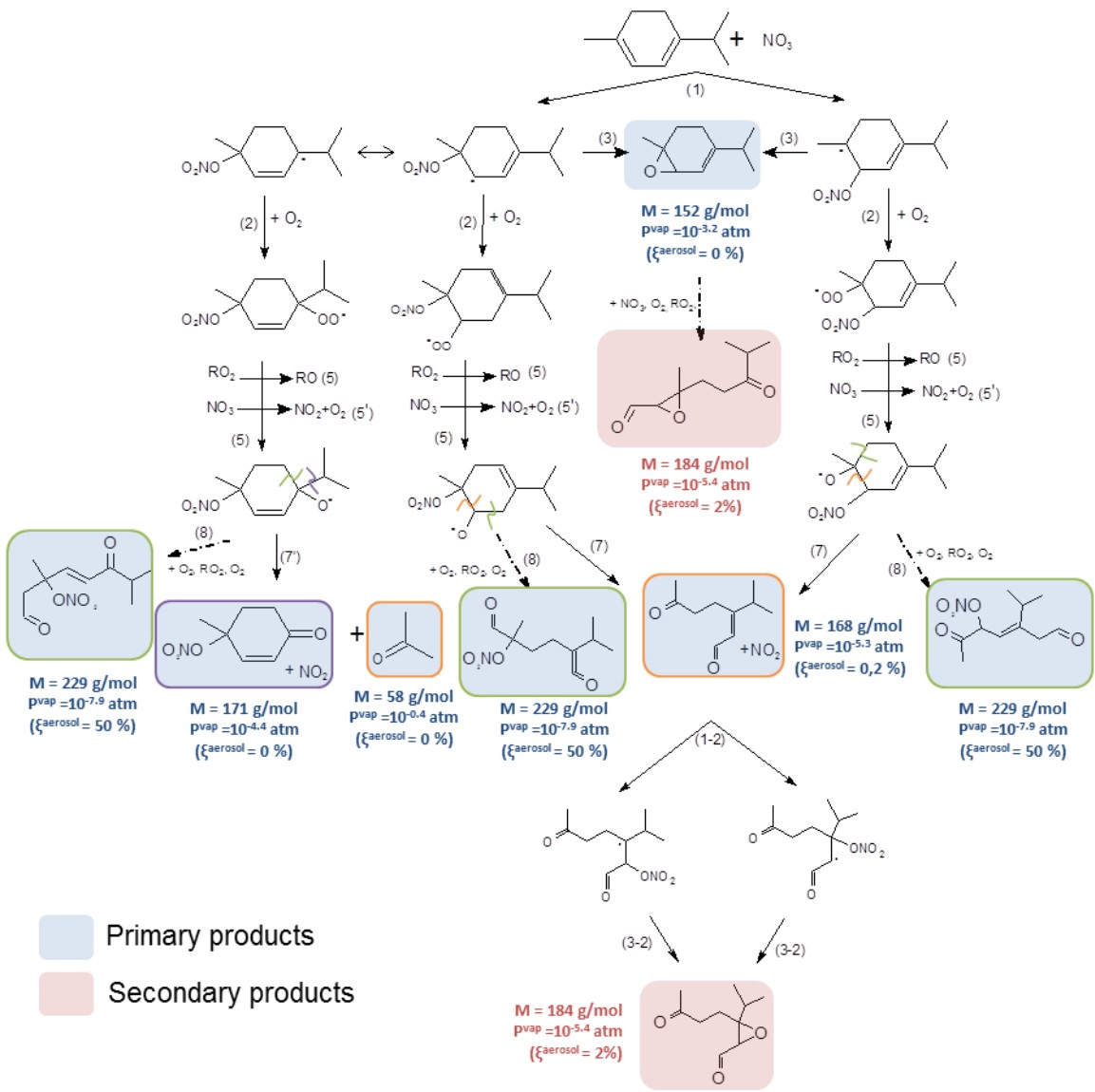


**Figure 8: Proposed mechanism for α-terpinene. First generation products are squared in blue and second generation ones in red. Alkoxy fragmentation products are squared according the location of the fragmentation. Molecular weight, vapor pressures and the gas/particle partition are shown next to the molecules. The reaction of the primary epoxide product is shown in the black square.**
