# Peer review of "Figure S1: Typical time profiles of an absolute rate determination experiment (2017/01/30) of signals measured by PTR-MS (lines) and FTIR (crosses)."

_Atmospheric Chemistry and Physics, 2020_

## Referee Comment (RC1) · Anonymous Referee #1 · 24 Jul 2020

General Comments

Fouqueau et. al. presents the results from a series of gas-phase reactions of the two monoterpenes $\alpha$-terpinene and $\gamma$-terpinene with the nitrate radical. They report rate constants obtained from absolute and relative methods as well as measurements of SOA and organic nitrate yields. While organonitrate yields were similar for each, $\gamma$-terpinene yielded significantly more SOA than $\alpha$-terpinene. Tentative product identifications using PTR-MS facilitated a discussion of potential reaction mechanisms for each system. The two monoterpenes are structurally similar, however, $\alpha$-terpinene possesses a conjugated system which gives rise to a radical mobility that is believed

to impact reactivity and contribute to suppression of SOA yields relative to $\gamma$-terpinene. Overall, the authors do a good job highlighting how a small change in chemical structure can have dramatic implications on chemical reactivity which need to be accounted for in modeling efforts.

The measurements and their discussion are appropriate for Atmospheric Chemistry and Physics and I recommend that it be accepted with minor revisions. I believe the data to be of high quality and an important contribution to the community's understanding of monoterpene oxidation reactions.

Specific Comments

I have two comments concerning the proposed mechanisms.

First, to facilitate a discussion of the proposed mechanisms the authors show only two radicals (line 440). I would have benefitted from seeing all four "options" drawn out for each species, even if the remaining two for each terpene were presented in the SI. The structures that formed are not necessarily identical. For example, $\gamma$ -terpinene oxidation (Figure 7, reaction 1, right branch), the authors show nitrate radical addition to the tertiary carbon which generates a secondary alkyl radical, when addition to the secondary carbon would yield a more stable tertiary radical. I appreciate that this is a complex mechanism and that the authors want to keep the figure as clean as possible, but something is missing by not showing all options.

Second, the alkoxy radical that would form on the tertiary carbon by the pathway described above would have a non-ring opening fragmentation pathway similar to what is shown in Figure 8, reaction 7' for $\alpha$-terpinene. On line 549 the authors discuss that ring openings are the most likely pathways based on the Vereecken and Peeters (2009) SAR and provide information on the energy, but ignore any discussion of the non-ring opening pathway. It would be helpful here to have some information on what the difference in energies and the expected branching ratios would be if the formation of the isopropyl radical were included. How different are the energies and would this lead

to the branching change at all? Their observations of relatively low concentrations of acetone are supported by this claim, but a discussion of why this might be the case would be helpful.

46. It is unclear if "highly soluble" refers to miscibility in the organic aerosol or aqueous phase.

85. The $\alpha$ -terpinene is only 85% pure and it is unclear if this factored into the any calculations, especially given the low SOA yields for this species.

151 What is the model of the Palas Welas used in this study? How was this data used in this study?

460. It is a personal preference to not suggest that signal intensity provide information on concentrations in the absence of the appropriate standards.

551. Branching ratios are unitless quantities but the sentence "with similar branching ratios (differing of 0.6 kcal/mole)" suggests that it has units of molar energy. If this is an activation energy, maybe state that.

Technical Corrections

56. The statement "trees emissions" should be "tree emissions"

133. The sentence containing "allowing to propose" is incomplete. Maybe allowing "us" to propose?

193. There is a typo in the parentheses in this sentence "products, RO2 radicals, . . .)"

248. It should be Raoult's Law.

890. The citation is missing page numbers.

900. The title has a typo. It reads "reaction barrier heightsw" but should be "heights".

Figure 7 and 8: The two different shades of gray used to differentiate Primary and Secondary Products are almost indistinguishable in my copy.

---

## Referee Comment (RC2) · Anonymous Referee #2 · 7 Aug 2020

**GENERAL COMMENTS**

In this manuscript the authors present results of an experimental study of the kinetics, products, and mechanisms of the reactions of NO3 radicals with two monoterpenes: a- and g-terpinene. Both compounds have two C=C double bonds, although in one case they are conjugated and in the other not. The study seeks to determine how these structural differences affect reaction rates, gas-phase products and mechanisms, and secondary organic aerosol (SOA) formation. Experiments were conducted in either a glass or stainless-steel chamber and gas phase products were monitored with a PTR-MS and in situ FTIR. SOA formation was monitored with an SMPS and composition

was probed by FTIR analysis of filter samples.

The relative and absolute rate constants measured for each monoterpene compare well with each other, with previously measured rate constants, and with predictions of structure-activity relationships. The authors present plausible gas-phase mechanisms to explain the formation of the proposed products identified by the PTR-MS, which include multifunctional compounds containing various combinations of nitrate, carbonyl, and hydroxyl groups. When combined with vapor pressure estimates the proposed products also explain the very large differences in the SOA yields measured for the two monoterpenes. Overall, this is a very comprehensive study and presents new results on the nighttime chemistry of monoterpenes that will be useful to the atmospheric chemistry community. I think the paper should be published in ACP after the following minor comments and have been addressed.

SPECIFIC COMMENTS

1. Line 86: The purity of the a-terpinene is not very high. Do the authors have any idea what else it contains? Are there any observed products that might not be explained by reaction of a-terpinene?

2. Line 335 and beyond: SOA yields are explained strictly in terms of physical partitioning of monomers from the gas phase to the particles. Claflin and Ziemann, J. Phys. Chem. A, 2018, showed that the SOA formed from the b-pinene + NO3 reaction consists almost entirely of acetal dimers formed in the particle phase. Could similar reactions occur in these systems, and how would the presence of oligomers alter the interpretation of SOA yields?

3. Line 335 and beyond: No mention is made as to the effects of loss of gas-phase products to the chamber walls. I would think this is quite high for stainless steel and possibly glass, and these sorts of products. This should be discussed.

4. Line 389: Could add SOA results from Claflin and Ziemann, J. Phys. Chem. A,

2018.

5. Line 468: Peroxynitrates can decompose to a carbonyl + HNO3 in the condensed phase. How might this affect the results?

TECHNICAL COMMENTS

1. Line 208: I think "et" should be "and".

2. Line 476: "bound" should be "bond".

---

## Author Comment (AC1) · 25 Sep 2020

First, the authors would like to thank the anonymous referee for this discussion and its constructive comments, corrections and suggestions that ensued. We have carefully replied to all its comments and the paper has been improved following his recommendations. All technical corrections suggested by the referee have been carefully performed. Answers have also been provided for all comments and changes have been performed accordingly. Please find below the answers to the comments:

Specific comments:

[Figure]

First, to facilitate a discussion of the proposed mechanisms the authors show only two radicals (line 440). I would have benefitted from seeing all four "options" drawn out for each species, even if the remaining two for each terpene were presented in the SI. The structures that formed are not necessarily identical. For example, $\gamma$-terpinene oxidation (Figure 7, reaction 1, right branch), the authors show nitrate radical addition to the tertiary carbon which generates a secondary alkyl radical, when addition to the secondary carbon would yield a more stable tertiary radical. I appreciate that this is a complex mechanism and that the authors want to keep the figure as clean as possible, but something is missing by not showing all options.

The mechanisms of the two other radicals for $\alpha$-terpinene and $\gamma$-terpinene have been added in the supporting information (Figures S4 & S5). Sentences have been added L. 456 ("The mechanism of the two other alkyl radicals is presented in Fig. S4.") and L. 538 ("The mechanism of the other two alkyl radicals is presented in Fig. S5.").

Second, the alkoxy radical that would form on the tertiary carbon by the pathway described above would have a non-ring opening fragmentation pathway similar to what is shown in Figure 8, reaction 7' for $\alpha$-terpinene. On line 549 the authors discuss that ring openings are the most likely pathways based on the Vereecken and Peeters (2009) SAR and provide information on the energy, but ignore any discussion of the non-ring opening pathway. It would be helpful here to have some information on what the difference in energies and the expected branching ratios would be if the formation of the isopropyl radical were included. How different are the energies and would this lead to the branching change at all? Their observations of relatively low concentrations of acetone are supported by this claim, but a discussion of why this might be the case would be helpful.

We indeed focused the discussion on ring opening fragmentation for the alkoxy radical. Using the SAR of Vereecken and Peeters (2009) the energy for the loss of the isopropyl or methyl group is calculated to be 10.2 kcal mol-1. It is the same for all the non-ring opening fragmentations, because the local configuration around the alkoxy group is the

same. As a reminder, the energy for the ring opening fragmentation is 6.0 and 6.5 kcal mol-1. A difference of 4 kcal mol-1 is significant enough to predict that this is a minor pathway. For $\gamma$-terpinene, no acetone formation has been detected and for $\alpha$-terpinene the acetone yield is very low, suggesting that the loss of the isopropyl group is a minor pathway, in good agreement with the SAR result. The loss of the methyl group leads to the formation of formaldehyde, which is not specific product. We decided not to discuss about.

Sentences have been added to complete the discussion: L. 476: "The SAR developed by Vereecken and Peeters, 2009 which is based on DFT calculations, has been used to estimate the energy barriers of the various reaction pathways of the alkoxy radicals. Energy barriers for reactions 7 and 8 appear to be similar (Eb,7 = 6.0 kcal mol-1 et Eb,8 = 6.5 kcal mol-1, with an error estimated by the authors of 0.5 kcal mol-1), leading to similar branching ratios for the two possible ring openings. The loss of methyl or isopropyl group presents an energy barrier significantly higher (10.2 kcal mol-1) whatever the alkoxy radical considered. These pathways appear minor compared to the ring opening and this is in good agreement with the fact that acetone was not detected during the experiments." L. 558: "The SAR proposed by Vereecken and Peeters, 2009 allowed estimating the energy barriers of the different evolution pathways of alkoxy radicals. It suggests that ring openings (reactions 7 and 8) are the two most likely pathways with similar energy barriers (respectively 6.0 and 6.5 kcal mol-1). As for -terpinene, reaction 7' corresponding to the loss of the isopropyl group appears to be less favorable with an energy barrier of 10.2 kcal mol-1."

46. It is unclear if "highly soluble" refers to miscibility in the organic aerosol or aqueous phase.

The "highly soluble" sentence refers to miscibility both in aqueous phase and in polar organic phase. This has been clarified in the manuscript: "both in aqueous phase and organic aerosol (Picquet-Varrault et al., 2019)" (L. 46). The following citations were added to references: Picquet-Varrault, B., Suarez, R., Duncianu, M., Cazaunau

M., Pangui, E., David, M. and Doussin, J.-F.; Photolysis and oxidation by OH radicals of two carbonyl nitrates: 4 nitrooxy-2-butanone and 5-nitrooxy 2-pentanone, Atmos. Chem. Phys., 20, 487–498, 2020.

85. The $\alpha$-terpinene is only 85% pure and it is unclear if this factored into the any calculations, especially given the low SOA yields for this species.

The referee is right when saying that the low purity of $\alpha$-terpinene may affect the organic nitrates and SOA yields. At the time the experiments were carried out, no higher purity sample was commercially available. We have contacted the company in order to have information about impurities but it was not able to provide it. Nevertheless, prior to each $\alpha$-terpinene injection, a purification stage was performed by pumping the sample in the vacuum line. This stage may contribute to remove high volatility impurities. For low volatility impurities, it is expected that they will remain in the sample (condensed phase). For yield calculations, it was therefore considered that only terpinene was introduced into the chamber. However, because these impurities remain unknown, we cannot state with certainty that this purification stage is 100% efficient. Therefore, it should be considered that this may generate additional uncertainty on yields which may be slightly underestimated. This has been explained in the manuscript L.86, L.341 and L. 411.

151 What is the model of the Palas Welas used in this study? How was this data used in this study?

The Palas Welas used is a Welas digital 2000. It has been used in the case of $\alpha$-terpinene in order to complete the distribution when the distribution measured by the SMPS was out of the measurement range. Information have been added in the article L. 154 ("conducted on $\alpha$-terpinene") and L. 155 ("Welas Digital 2000").

460. It is a personal preference to not suggest that signal intensity provide information on concentrations in the absence of the appropriate standards.

[Figure]

We agree that this hypothesis is debatable and we have removed it from the text. Following sentences have been removed: L. 469: "However, if one hypothesizes that the products have similar responses, one can consider that signal intensities provide information on major products and that a product which has a strong signal is largely formed." L463: "In both modes, the dicarbonyl compound coming from the decomposition of the RO radical by reaction 7 has an intense signal, suggesting a high formation yield." L.467: "The signal of the trifunctional compound formed by reaction 8 (m/z 230) is weak and can be explained by its low volatility which suggests a partition in favor of the aerosol phase."

551. Branching ratios are unitless quantities but the sentence "with similar branching ratios (differing of 0.6 kcal/mole)" suggests that it has units of molar energy. If this is an activation energy, maybe state that.

It is true the branching ratios are unitless and that we were referring to activation energies. It has been corrected L. 560.

Technical Corrections 56. The statement "trees emissions" should be "tree emissions" It has been corrected.

133. The sentence containing "allowing to propose" is incomplete. Maybe allowing "us" to propose? It has been corrected by "allowing the authors to propose".

193. There is a typo in the parentheses in this sentence "products, RO2 radicals,...)" It has been corrected.

248. It should be Raoult's Law. It has been corrected.

890. The citation is missing page numbers. It has been corrected.

900. The title has a typo. It reads "reaction barrier heightsw" but should be "heights". It has been corrected.

Figure 7 and 8: The two different shades of gray used to differentiate Primary and

Secondary Products are almost indistinguishable in my copy.

The mechanisms are not made with two different shades of gray used to differentiate Primary and Secondary Products but in pink and blue squares. These figures have been made to be seen with colors.

---

## Author Comment (AC2) · 25 Sep 2020

First, the authors would like to thank the anonymous referee for this discussion and its constructive comments, corrections and suggestions that ensued. We have carefully replied to all its comments and the paper has been improved following his recommendations. All technical corrections suggested by the referee have been carefully performed. Answers have also been provided for all comments and changes have been performed accordingly. Please find below the answers to the comments:

Specific comments

[Figure]

1. Line 86: The purity of the a-terpinene is not very high. Do the authors have any idea what else it contains? Are there any observed products that might not be explained by reaction of a-terpinene?

The referee is right when saying that the low purity of $\alpha$-terpinene may affect the organic nitrates and SOA yields. At the time the experiments were carried out, no higher purity sample was commercially available. We have contacted the company in order to have information about impurities but it was not able to provide it. . Nevertheless, prior to each $\alpha$-terpinene injection, a purification stage was performed by pumping the sample in the vacuum line. This stage may contribute to remove high volatility impurities. For low volatility impurities, it is expected that they will remain in the sample (condensed phase). For yield calculations, it was therefore considered that only terpinene was introduced into the chamber. However, because these impurities remain unknown, we cannot state with certainty that this purification stage is 100% efficient. Therefore, it should be considered that this may generate additional uncertainty on yields which may be slightly underestimated. This has been explained in the manuscript L.86, L.341 and L. 413.

2. Line 335 and beyond: SOA yields are explained strictly in terms of physical partitioning of monomers from the gas phase to the particles. Claflin and Ziemann, J. Phys. Chem. A, 2018, showed that the SOA formed from the $\beta$-pinene + NO3 reaction consists almost entirely of acetal dimers formed in the particle phase. Could similar reactions occur in these systems, and how would the presence of oligomers alter the interpretation of SOA yields? & 4. Line 389: Could add SOA results from Claflin and Ziemann, J. Phys. Chem. A, 2018.

This was an omission but we agree that we have to discuss this point. The reason of this omission was that no analysis at the molecular scale was conducted in the particle phase during our experiments. Indeed, in this study we only measure the total organic nitrates in the aerosol phase from their IR absorption band. If polymers are formed in the particle phase, for example acetal dimers and trimers (as presented by Claflin

and Ziemann, J. Phys. Chem. A, 2018), which have a nitrate group, they cannot be distinguished from the monomers in our analysis. The study of Claflin and Ziemann showed that hydroxynitrates can react withcarbonyl compounds, via an acid-catalyzed particle-phase reaction leading to the formation of acetal dimers and trimers. Our results are coherent with this statement: 1) For $\gamma$-terpinene, we identified hydroxynitrates and carbonyl nitrates in the gas phase, which have low enough volatility to transfer towards the particle phase. We are making the hypothesis that these molecules contribute the most to the SOA formation. Once in the particle phase, they can then react to form acetal dimers, but these last species can not detected in our study. 2) For $\alpha$-terpinene, we did not observe the formation of hydroxynitrate, thus the formation of these dimers is expected to be negligible. This is also in good agreement with the low SOA yields for this compound. A discussion has been added L. 612 to discuss about these results: "The study of Claflin and Ziemann, 2018 showed that the hydroxynitrates and the carbonyl compounds can react via an acid-catalyzed particle phase reaction leading to the formation of acetal dimers and trimers. No molecular analysis of the particle phase, except for the organic nitrates, was conducted. If polymers are formed in the particle phase, for example acetal dimers and trimers, which have a nitrate group, they cannot be distinguished from the monomers. For $\gamma$-terpinene, hydroxynitrates and carbonyl nitrates were detected in the gas phase, which have low enough volatility to go in particle phase and contribute the most to SOA formation. They can then react to form acetal dimers or trimers in the particle phase, but with no possible detection. For $\alpha$-terpinene, no hydroxynitrate formation was detected; the formation of these dimers is expected to be negligible. This is in good agreement with the low SOA yields for this compound. This study also showed the importance of RO2 + RO2 reaction and alkoxy decomposition, which are the key point in $\alpha$- and $\gamma$-terpinene chemistry. The importance of isomerization and acid-catalyzed particle phase reaction has not been proved but is coherent with the results."

The following citations were added to references: Claflin M. S. and Ziemann P. J.: Identification and Quantitation of Aerosol Products of the Reaction of $\beta$-Pinene with

NO3 Radicals and Implications for Gas- and Particle-Phase Reaction Mechanisms J. Phys. Chem. A, 122, 14, 3640–3652, 2018.

3. Line 335 and beyond: No mention is made as to the effects of loss of gas-phase products to the chamber walls. I would think this is quite high for stainless steel and possibly glass, and these sorts of products. This should be discussed.

The referee is right, organic nitrates are low volatile compounds, which can be lost on the walls. Mechanistic experiments were conducted in CESAM chamber, so with stainless steel walls, and it has been observed, from previous studies (Suarez-Bertoa et al., 2012; Picquet-Varrault et al., 2020), that organic nitrates adsorb on these walls. The loss rates of several multifunctional organic nitrates (in particular carbonyl-nitrates) have been studied and were found to range between 0.5-2 $\times$ 10-5 s-1. At the time scale of an experiment (3-4 hours), this may lead to a non-negligible loss of organic nitrates (up to 25%), and therefore to an underestimation of the yields, in particular of organic nitrates in the aerosol phase. However, yields of ONs in the gas-phase were calculated during the oxidation period, on a time scale of max. 1h and are therefore less affected by the wall losses (below 10%). The following text has been added: L. 379: "It is also expected that organic nitrates adsorb on the stainless steel walls. Indeed, the loss rates of several multifunctional organic nitrates (in particular carbonyl-nitrates) have been observed in previous studies (Suarez-Bertoa et al., 2012; Picquet-Varrault et al., 2020) and were found to range between 0.5 and 2 $\times$ 10-5 s-1. However, as yields of organic nitrates in the gas-phase were calculated during a relatively short period (less than 1 hour), these wall losses are expected to be low (less than 10%) and this is confirmed by the good linearity of the plots." And L. 408: "It should also be mentioned that these yields may be affected by wall losses of organic nitrates. Considering that these yields have been measured by collecting particles several hours after the beginning of the experiment, this may lead to a non-negligible loss of organic nitrates (up to 25%), and therefore to an underestimation of the yields. Because wall loss rates can vary from an experiment to another one, this can also explain the variability of YONp." The following

citations were added to references: Suarez-Bertoa, R., Picquet-Varrault, B., Tamas, W., Pangui, E., and Doussin, J.-F. ; Atmospheric Fate of a Series of Carbonyl Nitrates: Photolysis Frequencies and OH-Oxidation Rate Constants, Environ. Sci. Technol., 46, 22, 12502–12509, 2012. Picquet-Varrault, B., Suarez, R., Duncianu, M., Cazaunau M., Pangui, E., David, M. and Doussin, J.-F. ; Photolysis and oxidation by OH radicals of two carbonyl nitrates: 4 nitrooxy-2-butanone and 5-nitrooxy 2-pentanone, Atmos. Chem. Phys., 20, 487–498, 2020.

5. Line 468: Peroxynitrates can decompose to a carbonyl + HNO3 in the condensed phase. How might this affect the results?

We expect that these compounds are not formed. We indeed did not detect peroxynitrates both in the gas and particle phases, particularly using FTIR detection where these compounds have a very strong and distinctive signature. A precision has been added L.482: "It should also be noticed that peroxynitrates (RO2NO2), which have a characteristic absorption in the IR region, were not detected in our experiments, neither in the gaseous phase, nor in the aerosol one. This suggests that that RO2 + NO2 reactions are minor pathways."

Technical comments

1. Line 208: I think "et" should be "and". It has been corrected. 2. Line 476: "bound" should be "bond". It has been corrected.